# Robust and Decomposable Average Precision for Image Retrieval

**Elias Ramzi**[1,2]
elias.ramzi@cnam.fr

**Nicolas Thome**[1]
nicolas.thome@cnam.fr

**Clément Rambour**[1]
clement.rambour@cnam.fr

**Nicolas Audebert**[1]
nicolas.audebert@cnam.fr

**Xavier Bitot**[2]
xavier.bitot@coexya.eu

[1]CEDRIC, Conservatoire National des Arts et Métiers, Paris, France
[2]Coexya, Paris, France

## Abstract

In image retrieval, standard evaluation metrics rely on score ranking, e.g. average precision (AP). In this paper, we introduce a method for robust and decomposable average precision (ROADMAP) addressing two major challenges for end-to-end training of deep neural networks with AP: non-differentiability and non-decomposability. Firstly, we propose a new differentiable approximation of the rank function, which provides an upper bound of the AP loss and ensures robust training. Secondly, we design a simple yet effective loss function to reduce the decomposability gap between the AP in the whole training set and its averaged batch approximation, for which we provide theoretical guarantees. Extensive experiments conducted on three image retrieval datasets show that ROADMAP outperforms several recent AP approximation methods and highlight the importance of our two contributions. Finally, using ROADMAP for training deep models yields very good performances, outperforming state-of-the-art results on the three datasets. Code and instructions to reproduce our results will be made publicly available at https://github.com/elias-ramzi/ROADMAP.

## 1 Introduction

The task of 'query by example' is a major prediction problem, which consists in learning a similarity function able to properly rank all the instances in a retrieval set according to their relevance to the query, such that relevant items have the largest similarity. In computer vision, it drives several major applications, *e.g.* content-based image retrieval, face recognition or person re-identification.

Such tasks are usually evaluated with rank-based metrics, *e.g.* Recall@k, Normalized Discounted Cumulative Gain (NDCG), and Average Precision (AP). AP is also the *de facto* metric used in several vision tasks implying a large imbalance between positive and negative samples, *e.g.* object detection.

In this paper, we address the problem of direct AP training with stochastic gradient-based optimization, *e.g.* using deep neural networks, which poses two major challenges.

Firstly, the AP loss $\mathcal{L}_{AP} = 1 - AP$ is not differentiable and is thus not directly amenable to gradient-based optimization. There has been a rich literature for providing smooth and upper bound surrogate

35th Conference on Neural Information Processing Systems (NeurIPS 2021).

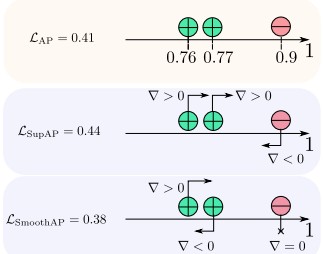
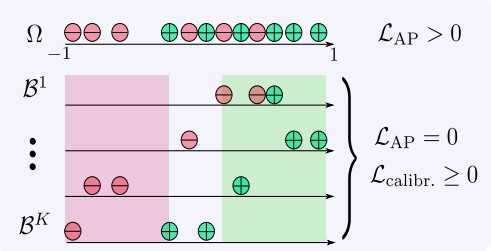

(a) $\mathcal{L}_{\text{SupAP}} \geq \mathcal{L}_{\text{AP}}$ and $\nabla \mathcal{L}_{\text{SupAP}} > 0$ in this example, in contrast to SmoothAP [2]. This ensures robust training and comes from a new approximation of the rank function.

(b) $\mathcal{L}_{\text{AP}}$ non-decomposability: $\mathcal{L}_{\text{AP}} = 0$ in all batches $\mathcal{B}^i$ despite $\mathcal{L}_{\text{AP}} \neq 0$ over the whole $\bigcup_i \mathcal{B}^i$. $\mathcal{L}_{\text{calibr.}}$ controls the absolute scores between batches, such that $\mathcal{L}_{\text{ROADMAP}} \neq 0$ in each batch.

Figure 1: Our robust and decomposable Average Precision training (ROADMAP) includes (a) a smooth loss $\mathcal{L}_{\text{SupAP}}$ upper-bounding $\mathcal{L}_{\text{AP}}$, and (b) a calibration loss $\mathcal{L}_{\text{calibr.}}$ supporting decomposability.

losses for $\mathcal{L}_{\text{AP}}$ [43, 21, 22, 6, 25]. More recently, smooth differentiable rank approximations have been proposed [35, 14, 15, 3, 27, 8, 2], but generally lose the important $\mathcal{L}_{\text{AP}}$ upper bound property.

The second important issue of AP optimization relates to its non-decomposability: $\mathcal{L}_{\text{AP}}^B$ averaged over batches underestimates $\mathcal{L}_{\text{AP}}$ on the whole training dataset, which we refer as the *decomposability gap*. In image retrieval, the attempts to circumvent the problem involve *ad hoc* methods based on batch sampling strategies [10, 32, 20, 32, 30], or storing all training representations/scores [39, 3, 27, 25], leading to complex models with a large computation and memory overhead.

In this paper, we introduce a method for RObust And DecoMposable Average Precision (ROADMAP), which explicitly addresses the aforementioned challenges of AP optimization.

Our first contribution is to propose a new surrogate loss $\mathcal{L}_{\text{SupAP}}$ for $\mathcal{L}_{\text{AP}}$. In particular, we introduce a smooth approximation of the rank function, with a different behaviour for positive and negative examples. By this design, $\mathcal{L}_{\text{SupAP}}$ provides an upper bound of $\mathcal{L}_{\text{AP}}$, and always back-propagates gradients when the correct ranking is not satisfied. These two features illustrated in the the toy example on Figure 1a are not fulfilled by binning approaches [3, 27] or by SmoothAP [2].

As a second contribution, we propose to improve the non-decomposability in AP training. To this end, we introduce a simple yet effective training objective $\mathcal{L}_{\text{calibr.}}$, which calibrates the scores among different batches by controlling the absolute value of positive and negative samples. We provide a theoretical analysis showing that $\mathcal{L}_{\text{calibr.}}$ decreases the decomposability gap. Figure 1 illustrates how $\mathcal{L}_{\text{calibr.}}$ can be leveraged to improve the overall ranking.

We provide a thorough experimental validation including three standard image retrieval datasets and show that ROADMAP outperforms state-of-the-art methods. We also report the large and consistent gain compared to rank/AP approximation baselines, and we highlight in the ablation studies the importance of our two contributions. Finally, ROADMAP does not entail any memory or computation overhead and remains competitive even with small batches.

## 2 Related work

We discuss here the literature in image retrieval dedicated to AP optimization, and compare to other approaches based on optimizing representations [23, 1, 44, 46, 33] in the experiments.

**Smooth AP approximations** Studying smooth surrogate losses for AP has a long history. The widely used surrogate for retrieval is to consider constraints based on pairs [41, 12, 26], triplets [11], quadruplets [18] or n-uplets [30] to enforce partial ranking. These metric learning methods optimize a very coarse upper bound on AP and need complex post-processing and tricks to be effective.

One option for training with AP is to design smooth upper bounds on the AP loss. Seminal works are based on structural SVMs [43, 21], with extensions to speed-up the "loss-augmented inference" [22] or to adapt to weak supervision [6]. Recently, a generic blackbox combinatorial solver has been introduced [25] and applied to AP optimization [28]. To overcome the brittleness of AP with respect to

small score variations, an *ad hoc* perturbation is applied to positive and negative scores during training. These methods provide elegant AP upper bounds, but generally are coarse AP approximations.

Other approaches rely on designing smooth approximations of the the rank function. This is done in soft-binning techniques [14, 15, 35, 3, 27] by using a smoothed discretization of similarity scores. Other approaches rely on explicitly approximating the non-differentiable rank functions using neural networks [8], or with a sum of sigmoid functions in the recent SmoothAP approach [2]. These approaches enable accurate AP approximations by providing tight and smooth approximations of the rank function. However, they do not guarantee that the resulting loss is an AP loss upper bound. The $\mathcal{L}_{\text{SupAP}}$ introduced in this work is based on a smooth approximation of the rank function leading to an upper bound on the AP loss, making our approach both accurate and robust.

**Decomposability in AP optimization** Batch training is mandatory in deep learning. However, the non-decomposability of AP is a severe issue, since it yields an inconsistent AP gradient estimator.

Non-decomposability is related to sampling informative constraints in simple AP surrogates, *e.g.* triplet losses, since the constraints' cardinality on the whole training set is prohibitive. This has been addressed by efficient batch sampling [13, 10, 32] or selecting informative constraints within mini-batches [30, 9, 4, 32]. In cross-batch memory technique [39], the authors assume a slow drift in learned representations to store them and compute global mining in pair-based deep metric learning.

In AP optimization, the non-decomposability has essentially been addressed by a brute force increase of the batch size [3, 27, 25]. This includes an important overhead in computation and memory, generally involving a two-step approach for first computing the AP loss and subsequently re-computing activations and back-propagating gradients. In contrast, our loss $\mathcal{L}_{\text{calibr.}}$ does not add any overhead and enables good performances for AP optimization even with small batches.

# 3   Robust and decomposable AP training

We present here our method for RObust And DecoMposable AP (ROADMAP) dedicated to direct optimization of a smooth surrogate of AP with stochastic gradient descent (SGD), see Fig. 2.

**Training context**  Let us consider a retrieval set $\Omega = \{\boldsymbol{x_j}\}_{j \in [\![1;N]\!]}$ composed of $N$ elements, and a set of $M$ queries included in $\Omega$, *i.e.* $\mathcal{Q} = \{\boldsymbol{q_i}\}_{i \in [\![1;M]\!]} \subseteq \Omega$. For each query $\boldsymbol{q_i}$, each element in $\Omega$ is assigned a label $y(\boldsymbol{x_j}, \boldsymbol{q_i}) \in \{+1; -1\}$, such that $y(\boldsymbol{x_j}, \boldsymbol{q_i}) = 1$ (resp. $y(\boldsymbol{x_j}, \boldsymbol{q_i}) = -1$) if $\boldsymbol{x_j}$ is relevant (resp. irrelevant) with respect to $\boldsymbol{q_i}$. This defines a query-dependent partitioning of $\Omega$ such that $\Omega = \mathcal{P}_i \cup \mathcal{N}_i$, where $\mathcal{P}_i := \{\boldsymbol{x_j} \in \Omega | y(\boldsymbol{x_j}, \boldsymbol{q_i}) = +1\}$ and $\mathcal{N}_i := \{\boldsymbol{x_j} \in \Omega | y(\boldsymbol{x_j}, \boldsymbol{q_i}) = -1\}$.

For each $\boldsymbol{x_j} \in \Omega$, we define a prediction model parametrized by parameters $\boldsymbol{\theta}$, *e.g.* a deep neural network, which provides a vectorial embedding $\mathbf{v}_{\boldsymbol{q_i}} \in \mathbb{R}^d$ of each element, *i.e.*: $\mathbf{v}_{\boldsymbol{q_i}} := f_{\boldsymbol{\theta}}(\boldsymbol{q_i})$. In the embedded space $\mathbb{R}^d$, we compute a similarity score between each query $\boldsymbol{q_i}$ and each element in $\Omega$, *e.g.* by using the cosine similarity: $s(\boldsymbol{q_i}, \boldsymbol{x_j}) = \frac{\mathbf{v}_{\boldsymbol{q_i}}{}^T \mathbf{v_j}}{||\mathbf{v_{q_i}}||^2 ||\mathbf{v_j}||^2}$.

During training, our goal is to optimize, for each query $\boldsymbol{q_i}$, the model parameters $\boldsymbol{\theta}$ such that positive elements are ranked before negatives. More precisely, we aim at minimizing the AP loss $\mathcal{L}_{\text{AP}_i}$ for each query $\boldsymbol{q_i}$ in the retrieval set $\Omega$. Our overall AP loss $\mathcal{L}_{\text{AP}}$ is averaged over all queries:

$$\mathcal{L}_{\text{AP}}(\boldsymbol{\theta}) = 1 - \frac{1}{M} \sum_{i=1}^{M} \text{AP}_i(\boldsymbol{\theta}), \quad \text{AP}_i(\boldsymbol{\theta}) = \frac{1}{|\mathcal{P}_i|} \sum_{k \in \mathcal{P}_i} \text{Pre}(k, \theta) = \frac{1}{|\mathcal{P}_i|} \sum_{k \in \mathcal{P}_i} \frac{\text{rank}^+(k, \theta)}{\text{rank}(k, \theta)} \quad (1)$$

where $\text{Pre}(k, \theta)$ is the precision for the $k^{\text{th}}$ positive example $\boldsymbol{x_k}$, $\text{rank}^+(k, \theta)$ its rank among positives $\mathcal{P}_i$, and the $\text{rank}(k, \theta)$ its rank over $\Omega = \mathcal{P}_i \cup \mathcal{N}_i$.

As previously mentioned, there are two main challenges with SGD optimization of AP in Eq. (1): i) $\text{AP}(\boldsymbol{\theta})$ is not differentiable with respect to $\boldsymbol{\theta}$, and ii) AP does not linearly decompose into batches. ROADMAP addresses both issues: we introduce the robust differentiable $\mathcal{L}_{\text{SupAP}}$ surrogate (Section 3.1), and add the $\mathcal{L}_{\text{calibr.}}$ loss (Section 3.2) to improve AP decomposability. Our final loss $\mathcal{L}_{\text{ROADMAP}}$ is a linear combination of $\mathcal{L}_{\text{SupAP}}$ and $\mathcal{L}_{\text{calibr.}}$, weighted by the hyperparameter $\lambda$:

$$\mathcal{L}_{\text{ROADMAP}}(\boldsymbol{\theta}) = (1 - \lambda) \cdot \mathcal{L}_{\text{SupAP}}(\boldsymbol{\theta}) + \lambda \cdot \mathcal{L}_{\text{calibr.}}(\boldsymbol{\theta}) \quad (2)$$

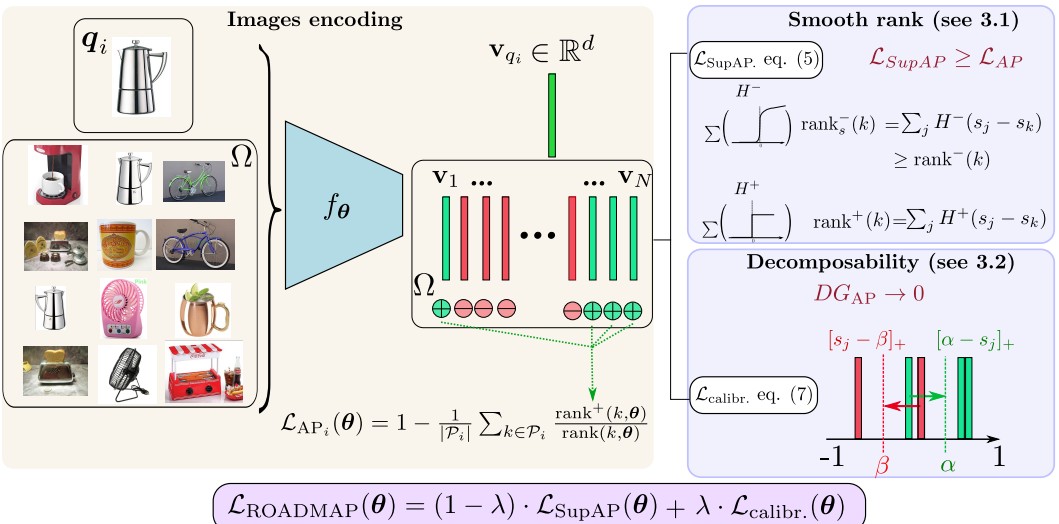

$$\mathcal{L}_{\text{ROADMAP}}(\boldsymbol{\theta}) = (1-\lambda) \cdot \mathcal{L}_{\text{SupAP}}(\boldsymbol{\theta}) + \lambda \cdot \mathcal{L}_{\text{calibr.}}(\boldsymbol{\theta})$$

Figure 2: ROADMAP training: we optimize parameters $\boldsymbol{\theta}$ of a deep neural networks to minimize a smooth surrogate of $\mathcal{L}_{\text{AP}_i}(\boldsymbol{\theta})$ between the query $\boldsymbol{q_i}$ and the retrieval set $\Omega$. Our smooth rank approximations $H^+$ and $H^-$ enables $\mathcal{L}_{\text{SupAP}}$ to be both accurate and robust (sec 3.1), and $\mathcal{L}_{\text{calibr.}}$ enables an implicit batch scores comparison for better decomposability without additional storing (sec 3.2).

## 3.1 Robustness in smooth rank approximation

The non-differentiablity in Eq (1) comes from the ranking operator, which can be viewed as counting the number of instances that have a similarity score greater than the considered instance, *i.e.*[1]:

$$\text{rank}^+(k) = 1 + \sum_{j \in \mathcal{P}_i \setminus \{k\}} H(s_j - s_k), \quad \text{where } H(t) = \begin{cases} 1 & \text{if } t \geq 0 \\ 0 & \text{otherwise} \end{cases}$$

$$\text{rank}(k) = \text{rank}^+(k) + \sum_{j \in \mathcal{N}_i} H(s_j - s_k) = \text{rank}^+(k) + \text{rank}^-(k) \tag{3}$$

From Eq. (3) it becomes clear that the non-differentiablity is due to the Heaviside (step) function $H$, whose derivative is either zero or undefined. Note that the computation of $\text{rank}^+(k)$ and $\text{rank}^-(k)$ in Eq. (3) relates to the rank of positive instances $\boldsymbol{x_k} \in \mathcal{P}_i$: the score $s_k$ in Eq. (3) is always the score of a positive, whereas $s_j$ can either be a negative's or positive's score.

**Smooth loss $\mathcal{L}_{\text{SupAP}}$** To provide a smooth approximation of $\mathcal{L}_{\text{AP}}$ in Eq. (1), we introduce a smooth approximation of the rank function. In particular, we propose a different behaviour between $\text{rank}^+(k)$ and $\text{rank}^-(k)$ in Eq. (3) by defining two functions $H^+$ and $H^-$.

For $\text{rank}^+(k)$, we choose to keep the Heaviside (step) function, *i.e.* $H^+ = H$ (see Fig. 3a), which consists in ignoring $\text{rank}^+(k)$ in gradient-based AP optimization. This is done on purpose since $\frac{\partial AP}{\partial \text{rank}^+(k)} = \frac{\text{rank}^-(k)}{(\text{rank}^+(k) + \text{rank}^-(k))^2} \geq 0$: the gradient would tend to increase $\text{rank}^+(k)$ and to decrease the score of $s_k$. Reminding $\boldsymbol{x_k}$ is always a positive instance, this behaviour is undesirable.

For $\text{rank}^-(k)$, we define the following smooth surrogate $H^-$ for $H$, shown in Fig 3b:

$$H^-(t) = \begin{cases} \sigma(\frac{t}{\tau}) & \text{if } t \leq 0, \quad \text{where } \sigma \text{ is the sigmoid function (Fig. 3c)} \\ \sigma(\frac{t}{\tau}) + 0.5 & \text{if } t \in [0; \delta] \quad \text{with } \delta \geq 0 \\ \rho \cdot (t - \delta) + \sigma(\frac{\delta}{\tau}) + 0.5 & \text{if } t > \delta \end{cases} \tag{4}$$

where $\tau$ and $\rho$ are hyperparameters, and $\delta$ is defined such that the sigmoidal part of $H^-$ reaches the saturation regime and is fixed for the rest of the paper (see supplementary Sec. A). From

---

[1]For the sake of readability we drop in the following the dependence on $\boldsymbol{\theta}$ for the rank, *i.e.* $\text{rank}(k) := \text{rank}(k, \theta)$ and on the query for the similarity, *i.e.* $s_j := s(q_i, x_j)$.

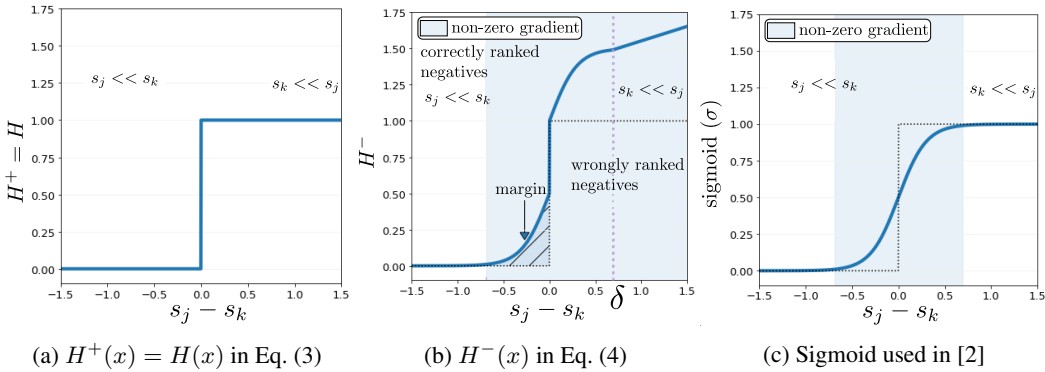

(a) $H^+(x) = H(x)$ in Eq. (3)      (b) $H^-(x)$ in Eq. (4)      (c) Sigmoid used in [2]

Figure 3: Proposed surrogate losses for the Heaviside (step): with $H^+(x)$ in Fig. 3a and $H^-(x)$ in Fig. 3b, $\mathcal{L}_{\text{SupAP}}$ in Eq. (5) is an upper bound of $\mathcal{L}_{\text{AP}}$. In addition, $H^-(x)$ back-propagates gradients until the correct ranking is satisfied, in contrast to the sigmoid used in [2] (Fig. 3c).

the $H^-$ smooth approximation defined in Eq. (4), we obtain the following smooth approximation $\text{rank}_s^-(k) = \sum_{j \in \mathcal{N}_i} H^-(s_j - s_k)$, leading to the following smooth AP loss approximation:

$$\mathcal{L}_{\text{SupAP}}(\boldsymbol{\theta}) = 1 - \frac{1}{M} \sum_{i=1}^{M} \frac{1}{|\mathcal{P}_i|} \sum_{k \in \mathcal{P}_i} \frac{\text{rank}^+(k)}{\text{rank}^+(k) + \text{rank}_s^-(k)} \tag{5}$$

**$\mathcal{L}_{\text{SupAP}}$ in Eq. (5) fulfills two main features for AP optimization:**

▶ ① **$\mathcal{L}_{\text{SupAP}}$ is an upper bound of $\mathcal{L}_{\text{AP}}$ in Eq. (1).** Since $H^-$ in Eq. (4) is an upper bound of a step function (Fig 3b), it is easy to see that $\mathcal{L}_{SupAP} \geq \mathcal{L}_{\text{AP}}$. This is a very important property, since it ensures that the model keeps training until the correct ranking is obtained. It is worth noting that existing smooth rank approximations in the literature [35, 3, 27, 2] do not fulfill this property.

▶ ② **$\mathcal{L}_{\text{SupAP}}$ brings training gradients until the correct ranking plus a margin is fulfilled.** When the ranking is incorrect, the negative $\boldsymbol{x_j}$ is ranked before the positive $\boldsymbol{x_k}$, thus $s_j > s_k$ and $H^-(s_j - s_k)$ in Eq. (4) has a non-null derivative. We use a sigmoid to have a large gradient when $s_j - s_k$ is small. To overcome vanishing gradients of the sigmoid for large values $s_j - s_k$, we use a linear function ensuring constant $\rho$ derivative. When the ranking is correct ($s_j < s_k$), we enforce robustness by imposing a margin parametrized by $\tau$ (sigmoid in Eq. (4)). This margin overcomes the brittleness of rank losses, which vanish as soon as the ranking is correct [14, 3, 25].

**Comparison to SmoothAP [2]** $\mathcal{L}_{\text{SupAP}}$ differs from $\mathcal{L}_{\text{SmoothAP}}$ in [2] by i) providing an upper bound on $\mathcal{L}_{\text{AP}}$, ii) improving the gradient flow (Fig. 3b vs Fig. 3c), and iii) overcoming adverse effects of the sigmoid for $rank^+$, as shown in Fig. 1a (and in supplementary sec. A). We experimentally verify the consistent gain brought out by $\mathcal{L}_{\text{SupAP}}$ over $\mathcal{L}_{\text{SmoothAP}}$.

### 3.2 Decomposable Average Precision

In Eq. (1), AP decomposes linearly between queries $\boldsymbol{q_i}$, but $\text{AP}_i$ does not decomposes linearly between samples. We therefore focus our analysis of the non-decomposability on a single query. For a retrieval set $\Omega$ of $N$ elements, we consider $\{\mathcal{B}^b\}_{b \in \{1:K\}}$ batches of size B, such that $N/B = K \in \mathbb{N}$. Let $\text{AP}_i^b(\boldsymbol{\theta})$ be the AP in batch $b$ for query $\boldsymbol{q_i}$, we define the "decomposability gap" $DG_{\text{AP}}$ as follows:

$$DG_{\text{AP}}(\boldsymbol{\theta}) = \frac{1}{K} \sum_{b=1}^{K} \text{AP}_i^b(\boldsymbol{\theta}) - \text{AP}_i(\boldsymbol{\theta}) \tag{6}$$

$DG_{\text{AP}}$ in Eq. (6) is a direct measure of the non-decomposability of AP (see supplementary Sec. A). Our motivation here is to decrease $DG_{\text{AP}}$, *i.e.* to have the average AP over the batches as close as possible to the AP computed over the whole training set. To this aim, we introduce the following loss

during training:

$$\mathcal{L}_{\text{calibr.}}(\boldsymbol{\theta}) = \frac{1}{M}\sum_{i=1}^{M}\underbrace{\frac{1}{|\mathcal{P}_i|}\sum_{\boldsymbol{x_j}\in\mathcal{P}_i}[\alpha - s_j]_+}_{\mathcal{L}_{\text{calibr.}}^{+}} + \underbrace{\frac{1}{|\mathcal{N}_i|}\sum_{\boldsymbol{x_j}\in\mathcal{N}_i}[s_j - \beta]_+}_{\mathcal{L}_{\text{calibr.}}^{-}} \tag{7}$$

where $[x]_+ = \max(0, x)$. The loss $\mathcal{L}_{\text{calibr.}}^{+}$ enforces the score of the positive $\boldsymbol{x_i}\in\mathcal{P}_i$ to be larger than $\alpha$, and $\mathcal{L}_{\text{calibr.}}^{-}$ enforces the score of the negative $\boldsymbol{x_j}\in\mathcal{N}_i$ to be smaller than $\beta < \alpha$. $\mathcal{L}_{\text{calibr.}}$ is a standard pair-based loss [12], which we revisit in our context to "calibrate" the values of the scores between mini-batches: intuitively, the fact that the positive (resp. negative) scores are above (resp. below) a threshold in the mini-batches makes the average AP closer to the AP on the whole dataset.

**Upper bound on the decomposabilty gap** To formalize this idea, we provide a theoretical analysis of the impact on the global ranking of $\mathcal{L}_{\text{calibr.}}$ in Eq. (7). Firstly, we can see that if $\mathcal{L}_{\text{calibr.}}^{-} = \mathcal{L}_{\text{calibr.}}^{+} = 0$, on each batch, the overall AP and the AP in batches is null, *i.e.* $DG_{\text{AP}}(\boldsymbol{\theta}) = 0$ and we get a decomposable AP. In a more general setting, we show that minimizing $\mathcal{L}_{\text{calibr.}}$ on each batch reduces the decomposability gap, hence improving the decomposability of the AP.

Let's consider $K$ batches $\{\mathcal{B}^b\}_{b\in\{1:K\}}$ of batch size $B$ divided in $\mathcal{P}_i^b$ positive instances and $\mathcal{N}_i^b$ negative instances w.r.t. the query $\boldsymbol{q_i}$. To give some insight we assume that the AP of each batch is one (*i.e.* $AP_i^b = 1$), and give the following upper bound of $DG_{\text{AP}}$ :

$$0 \le DG_{\text{AP}} \le 1 - \frac{1}{\sum_{b=1}^{K}|\mathcal{P}_i^b|}\left(\sum_{b=1}^{K}\sum_{j=1}^{B}\frac{j + |\mathcal{P}_i^1| + \cdots + |\mathcal{P}_i^{b-1}|}{j + |\mathcal{P}_i^1| + \cdots + |\mathcal{P}_i^{b-1}| + |\mathcal{N}_i^1| + \cdots + |\mathcal{N}_i^{b-1}|}\right) \tag{8}$$

This upper bound of the decomposability gap is given in the worst case for the global AP : the global ranking is built from the juxtaposition of the batches (see supplementary Sec. A).

We can refine this upper bound by introducing the calibration loss $\mathcal{L}_{\text{calibr.}}$ and constraining the scores of positive and negative instances to be well calibrated. On each batch we define the following quantities $E_b^- = \sum_{j\in\mathcal{N}_i^-}\mathbb{1}(s_j > \beta)$ which are the negative instances that do not respect the constraints and $G_b^- = \sum_{j\in\mathcal{N}_i^-}\mathbb{1}(s_j \le \beta)$ the negative instances that do. We similarly define $E_b^+$ and $G_b^+$. We then have the following upper bound on the decomposability gap :

$$0 \le DG_{\text{AP}} \le 1 - \frac{1}{\sum_{b=1}^{K}|\mathcal{P}_i^b|}\left(\sum_{b=1}^{K}\left[\sum_{j=1}^{G_b^+}\frac{j + G_1^+ + \cdots + G_{b-1}^+}{j + G_1^+ + \cdots + G_{b-1}^+ + E_1^- + \ldots E_{b-1}^-} + \right.\right. \tag{9}$$

$$\left.\left. \sum_{j=1}^{E_b^+}\frac{j + G_b^+ + |\mathcal{P}_i^1| + \cdots + |\mathcal{P}_i^{b-1}|}{j + G_b^+ + |\mathcal{P}_i^1| + \cdots + |\mathcal{P}_i^{b-1}| + |\mathcal{N}_i^1| + \cdots + |\mathcal{N}_i^{b-1}|}\right]\right)$$

This refined upper bound is tighter than the upper bound of Eq. (8). Our new $\mathcal{L}_{\text{calibr.}}$ loss directly optimizes this upper bound (by explicitly optimizing $E_b^-, E_b^+, E_b^+, G_b^+$), making it tighter, hence improving the decomposability of the AP (see supplementary Sec. A).

## 4 Experiments

**Experimental setup** We evaluate ROADMAP on the following three image retrieval datasets:
**CUB-200-2011** [37] contains 11 788 images of birds classified into 200 fine-grained classes. We follow the standard protocol and use the first (resp. last) 100 classes for training (resp. evaluation).
**Stanford Online Product (SOP)** [31] is a dataset with 120 053 images of 22 634 objects classified into 12 categories (*e.g.* bikes, coffee makers). We use the reference train and test splits from [31].
**INaturalist-2018** [36] is a large scale dataset of 461 939 wildlife animals images classified into 8142 classes. We use the splits from [2] with 70% of the classes in the train set and the rest in the test set.

**ROADMAP settings** For all experiments in Section 4.1 and Section 4.2, we use $\lambda = 0.5$ for $\mathcal{L}_{\text{ROADMAP}}$ in Eq. (2), $\tau = 0.01$ and $\rho = 100$ for $\mathcal{L}_{\text{SupAP}}$ in Eq. (5), $\alpha = 0.9$ and $\beta = 0.6$ for $\mathcal{L}_{\text{calibr.}}$

in Eq. (7). We study more in depth the impact of those parameters in Section 4.3. Deep models are trained using Adam [17] for ResNet-50 backbones and AdamW [19] for DeiT transformers [34].

**Test protocol** Methods are evaluated using the standard recall at k (R@k) and mean average precision at R [24] (mAP@R) metrics (see supplementary Sec. B).

## 4.1 ROADMAP validation

In this section, all models are trained in the same setting (ResNet-50 backbone, embedding size 512, batch size 64). The comparisons thus directly measures the impact of the training loss.

**Comparison to AP approximations.** In Table 1, we compare ROADMAP on the three datasets to recent AP loss approximations including the soft-binning approaches FastAP [3] and SoftBinAP [27], the generic solver BlackBox [28], and the smooth rank approximation [2]. We use the publicly available PyTorch implementations of all these baselines. We can see that ROADMAP outperforms all the current AP approximations by a large margin. The gain is especially pronounced on the large scale dataset INaturalist. This highlights the importance our two contributions, *i.e.* our robust smooth AP upper bound and our AP decomposability improvement (see supplementary Sec. B).

Table 1: Comparison between ROADMAP and state-of-the-art AP ranking based methods.

|  | CUB | | SOP | | INaturalist | |
| --- | --- | --- | --- | --- | --- | --- |
| Method | R@1 | mAP@R | R@1 | mAP@R | R@1 | mAP@R |
| FastAP [3] | 58.9 | 22.9 | 78.2 | 51.3 | 53.5 | 19.6 |
| SoftBin [27] | 61.2 | 24.0 | 80.1 | 53.5 | 56.6 | 20.1 |
| BlackBox [28] | 62.6 | 23.9 | 80.0 | 53.1 | 52.3 | 15.2 |
| SmoothAP [2] | 62.1 | 23.9 | 80.9 | 54.6 | 59.8 | 20.7 |
| ROADMAP | **64.2** | **25.3** | **82.0** | **56.5** | **64.5** | **25.1** |

**Comparison to memory methods.**

XBM stores the embeddings of previously seen batches to alleviate complex batch sampling and better approximate AP on the whole dataset. Although XBM has a low memory overhead (a few hundreds megabytes on SOP), it is time consuming. We ran experiments storing the entire dataset for SOP (60k embeddings), but for INaturalist we could not train while storing all the dataset in tractable time. We chose to store the same amount of embeddings as for SOP : 60k embeddings which is about 17% of the training set.

We can see in Table 2 that XBM is approximately 3 times longer to train than ROADMAP. This becomes critical on INaturalist, where training while storing 60k images takes about 3 days, and reaches only a R@1 of 60. Consequently, ROADMAP outperforms XBM on both datasets; there is a ∼+2pt increase on both metrics for SOP and an especially large gap on INaturalist. In the latter, not being able to store all the embeddings affects drastically the performances of the XBM in a negative way. There is a 5pt difference in R@1 and more than 6pt in mAP@R. This demonstrates the suitability of ROADMAP on large-scale settings.

Table 2: Our method compared to cross batch memory [39]. The unit of time is m/e which stands for minutes per epoch.

|  | SOP | | | INaturalist | | |
| --- | --- | --- | --- | --- | --- | --- |
| Method | R@1 | mAP@R | time↓ | R@1 | mAP@R | time↓ |
| XBM [39] | 80.6 | 54.9 | 6 | 59.3 | 18.5 | 34 |
| ROADMAP (ours) | **82.0** | **56.5** | **2** | **64.5** | **25.1** | **12** |

**Ablation study.** To study more in depth the impact of our contributions, we perform ablation studies in Table 3. We show the improvement against SmoothAP [2] when changing the sigmoid by $H^+$ and $H^-$ for $\mathcal{L}_{\text{SupAP}}$ in Eq. (5), and the use of $\mathcal{L}_{\text{calibr.}}$ in Eq. (7). We can see that $\mathcal{L}_{\text{SupAP}}$ consistently improves performances over $\mathcal{L}_{\text{SmoothAP}}$ (0.9pt on CUB, 0.5pt on SOP and 1.5pt on INaturalist). $\mathcal{L}_{\text{SupAP}}$ and $\mathcal{L}_{\text{calibr.}}$ equally contribute to the overall gain in CUB and SOP, but the gain of $\mathcal{L}_{\text{calibr.}}$ is much

more important on INaturalist. This is explained by the fact that the batch vs. dataset ratio size $\frac{B}{N}$ is tiny ($\ll 1$), making the decomposability gap in Eq. (6) huge. We can see that $\mathcal{L}_{\text{calibr.}}$ is very effective for reducing this gap and brings a gain of more than 3pt.

Table 3: Ablation study for the impact of our two contribution on and the SmoothAP baseline.

| Method | $H^-$ | $\mathcal{L}_{\text{calibr.}}$ | CUB R@1 | CUB mAP@R | SOP R@1 | SOP mAP@R | INaturalist R@1 | INaturalist mAP@R |
|---|---|---|---|---|---|---|---|---|
| SmoothAP [2] | ✗ | ✗ | 62.1 | 23.9 | 80.9 | 54.6 | 59.7 | 20.7 |
| SupAP | ✓ | ✗ | 62.9 | 24.6 | 81.4 | 55.3 | 61.2 | 21.3 |
| ROADMAP | ✓ | ✓ | **64.2** | **25.3** | **82.0** | **56.5** | **64.5** | **25.1** |

## 4.2 State of the art comparison

We compare ROADMAP to other state of the art methods across three image retrieval datasets and report the results in Table 4. We divide competitor methods into three categories: metric learning [29, 38, 45, 16, 39, 42], classification losses for image retrieval [46, 44, 1, 33], and AP approximations [3, 28, 2]. ROADMAP falls in the latter category. We use the same setup as in Section 4.1 and follow standard practices for ResNet-50 [33, 42, 1] by using larger images ($256 \times 256$ on SOP and CUB) and using max instead of average pooling and layer normalization for CUB.

Using the popular ResNet-50 backbone, ROADMAP establishes a new state of the art across all methods for SOP and the challenging INaturalist dataset and outperforms all previous AP approximations on CUB, while being competitive with the other two top performers (ProxyNCA++ and SEC). R@k improvements are consistent on all datasets with a $\sim$2pts R@1 increase on INaturalist and $\sim$3pts increase on SOP compared to SmoothAP, the best performing AP approximation from the literature.

Switching the backbone to the more recent vision transformer architecture DeiT [5, 34], further lifts the performances of ROADMAP by several point, from 3 to 9 points depending on the dataset, with a smaller embedding size (384 $vs$ 512). The decomposable AP approximation ROADMAP also outperforms by a significant margin IRT$_\text{R}$, the DeiT architecture for image retrieval introduced in [7] trained with a contrastive loss. Overall ROADMAP achieves state-of-the-art performances across all three datasets by a significant margin.

## 4.3 Model Analysis

We show in Fig. 4 the impact of the main ROADMAP hyperparameters on INaturalist. The relative weighting $\lambda$ from Eq. (2) controls the balance between our two training objectives $\mathcal{L}_{\text{SupAP}}$ and $\mathcal{L}_{\text{calibr.}}$: $\lambda = 0$ reduces $\mathcal{L}_{\text{ROADMAP}}$ to $\mathcal{L}_{\text{SupAP}}$ while $\lambda = 1$ to $\mathcal{L}_{\text{calibr.}}$. We can see in Fig. 4a that training with the complete $\mathcal{L}_{\text{ROADMAP}}$ with both $\mathcal{L}_{\text{calibr.}}$ and $\mathcal{L}_{\text{SupAP}}$ is always better than using only one of the two losses. Note that results are stable in the $[0.2, 0.8]$ range with a consistent $\sim$1.5pt increase, demonstrating the robustness of ROADMAP to this hyperparameter tuning.

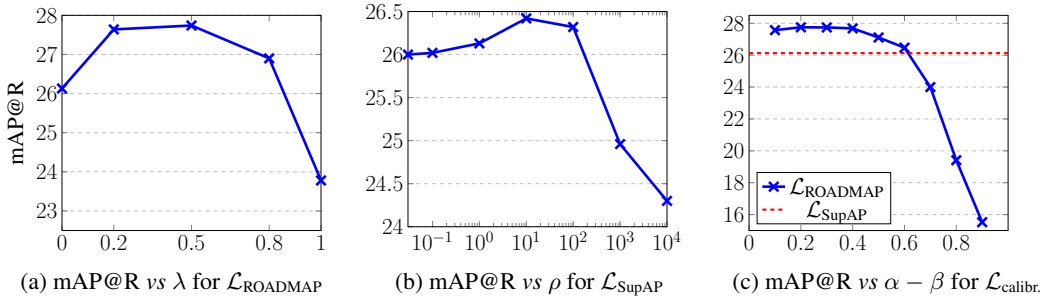

(a) mAP@R $vs$ $\lambda$ for $\mathcal{L}_{\text{ROADMAP}}$     (b) mAP@R $vs$ $\rho$ for $\mathcal{L}_{\text{SupAP}}$     (c) mAP@R $vs$ $\alpha - \beta$ for $\mathcal{L}_{\text{calibr.}}$

Figure 4: Analysis of ROADMAP hyperparameters on INaturalist (batch size 224).

Fig. 4b shows the influence of the slope $\rho$ that controls the linear regime in $H^-$ and determines the amount of gradient backpropagated for negative samples with a (wrong) high score. As shown in

Table 4: Comparison of state of the art performances from the literature on SOP, CUB and INaturalist with the proposed ROADMAP (recall@k). Except for the DeiT category, all methods rely on a standard convolutional backbone (generally ResNet-50).

| | Method | dim | SOP 1 | 10 | 100 | CUB 1 | 2 | 4 | 8 | INaturalist 1 | 4 | 16 | 32 |
|---|---|---|---|---|---|---|---|---|---|---|---|---|---|
| Metric learning | Triplet SH [40] | 512 | 72.7 | 86.2 | 93.8 | 63.6 | 74.4 | 83.1 | 90.0 | 58.1 | 75.5 | 86.8 | 90.7 |
| | LiftedStruct [31] | 512 | 62.1 | 79.8 | 91.3 | 47.2 | 58.9 | 70.2 | 80.2 | - | - | - | - |
| | MIC [29] | 512 | 77.2 | 89.4 | 95.6 | 66.1 | 76.8 | 85.6 | - | - | - | - | - |
| | MS [38] | 512 | 78.2 | 90.5 | 96.0 | 65.7 | 77.0 | 86.3 | 91.2 | - | - | - | - |
| | SEC [45] | 512 | 78.7 | 90.8 | 96.6 | 68.8 | 79.4 | 87.2 | 92.5 | - | - | - | - |
| | HORDE [16] | 512 | 80.1 | 91.3 | 96.2 | 66.8 | 77.4 | 85.1 | 91.0 | - | - | - | - |
| | XBM [39] | 128 | 80.6 | 91.6 | 96.2 | 65.8 | 75.9 | 84.0 | 89.9 | - | - | - | - |
| | Triplet SCT [42] | 512/64 | 81.9 | 92.6 | 96.8 | 57.7 | 69.8 | 79.6 | 87.0 | - | - | - | - |
| Classification | ProxyNCA [23] | 512 | 73.7 | - | - | 49.2 | 61.9 | 67.9 | 72.4 | 61.6 | 77.4 | 87.0 | 90.6 |
| | ProxyGML [46] | 512 | 78.0 | 90.6 | 96.2 | 66.6 | 77.6 | 86.4 | - | - | - | - | - |
| | NSoftmax [44] | 512 | 78.2 | 90.6 | 96.2 | 61.3 | 73.9 | 83.5 | 90.0 | - | - | - | - |
| | NSoftmax [44] | 2048 | 79.5 | 91.5 | 96.7 | 65.3 | 76.7 | 85.4 | 91.8 | - | - | - | - |
| | Cross-Entropy [1] | 2048 | 81.1 | 91.7 | 96.3 | 69.2 | 79.2 | 86.9 | 91.6 | - | - | - | - |
| | ProxyNCA++ [33] | 512 | 80.7 | 92.0 | 96.7 | 69.0 | 79.8 | 87.3 | 92.7 | - | - | - | - |
| | ProxyNCA++ [33] | 2048 | 81.4 | 92.4 | 96.9 | 72.2 | 82.0 | 89.2 | 93.5 | - | - | - | - |
| AP loss | FastAP [3] | 512 | 76.4 | 89.0 | 95.1 | - | - | - | - | 60.6 | 77.0 | 87.2 | 90.6 |
| | BlackBox [28] | 512 | 78.6 | 90.5 | 96.0 | 64.0 | 75.3 | 84.1 | 90.6 | 62.9 | 79.4 | 88.7 | 91.7 |
| | SmoothAP [2] | 512 | 80.1 | 91.5 | 96.6 | - | - | - | - | 67.2 | 81.8 | 90.3 | 93.1 |
| | SoftBin* [27] | 512 | 80.6 | 91.3 | 96.1 | 61.2 | 73.14 | 83.0 | 89.5 | 64.2 | 77.1 | 82.7 | 91.7 |
| | ROADMAP (ours) | 512 | 83.1 | 92.7 | 96.3 | 68.5 | 78.7 | 86.6 | 91.9 | 69.1 | 83.1 | 91.3 | 93.9 |
| DeiT | IRT$_R$ [7] | 384 | 84.2 | 93.7 | 97.3 | 76.6 | 85.0 | 91.1 | 94.3 | - | - | - | - |
| | ROADMAP (ours) | 384 | **86.0** | **94.4** | **97.6** | **77.4** | **85.5** | **91.4** | **95.0** | **73.6** | **86.2** | **93.1** | **95.2** |

Fig. 4b, the improvement is important and stable in $[10, 100]$. Note that $\rho > 0$ already improves the results compared to $\rho = 0$ in [2]. There is an important decrease when $\rho \gg 100$ probably due to the high gradient that takes over the signal for correctly ranked samples.

The impact of the margin $\alpha - \beta$ in $\mathcal{L}_{\text{calibr.}}$ is shown in Fig. 4c. Once again, ROADMAP exhibits a robust behaviour w.r.t. the values of its hyperparameters: any margin in the $[0.1, 0.6]$ range results in an improvement in mAP@R compared to the $\mathcal{L}_{\text{SupAP}}$ baseline without the decomposability loss. Best results are achieved with smaller margins $0.1 < \alpha - \beta < 0.4$.

Fig. 5 shows the improvement in mAP@R on the three datasets when adding $\mathcal{L}_{\text{calibr.}}$ to $\mathcal{L}_{\text{SupAP}}$. We can see that the increase becomes larger as the batch size gets smaller. This confirms our intuition that the decomposability in $\mathcal{L}_{\text{calibr.}}$ has a stronger effect on smaller batch sizes, for which the AP estimation is noisier and $DG_{\text{AP}}$ larger. This is critical on the large-scale dataset INaturalist where the batch AP on usual batch sizes is a very poor approximation of the global AP.

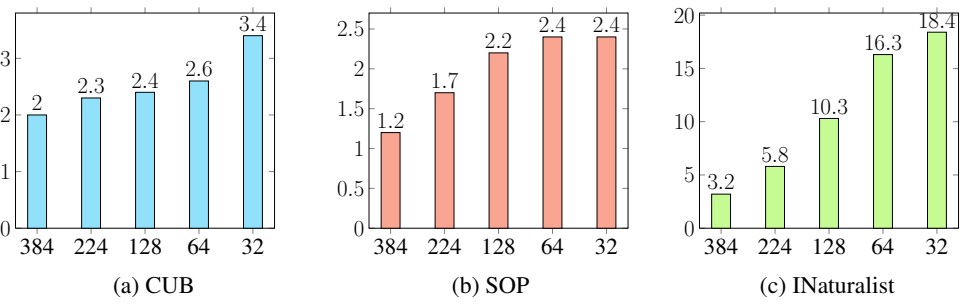

Figure 5: Relative increase of the mAP@R *vs* batch size when adding $\mathcal{L}_{\text{calibr.}}$ to $\mathcal{L}_{\text{SupAP}}$.

As a qualitative assessment, we show in Fig. 6 some results of ROADMAP on INaturalist. We show the queries (in purple) and the 4 most similar retrieved images (in green). We can appreciate the semantic quality of the retrieval. More qualitative results are provided in supplementary Sec. C.

Fig. 7 shows another qualitative assessment on INaturalist, where ROADMAP corrects some failing cases of the SmoothAP baseline.

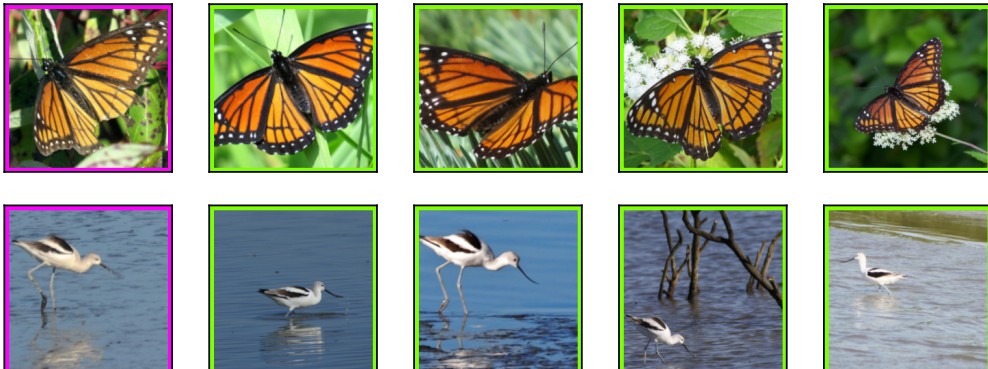

Figure 6: Results on INaturalist: a query (purple) with the 4 most similar retrieved images (green).

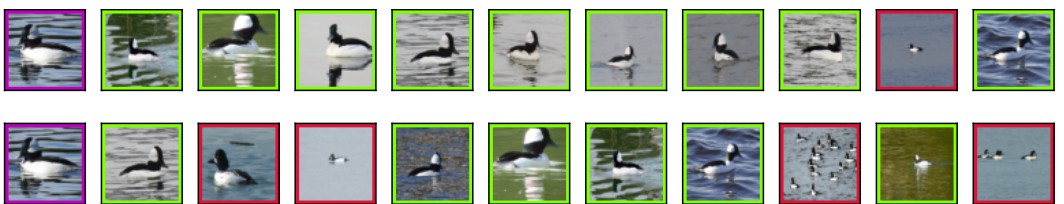

Figure 7: Results on INaturalist: a query (purple) with the 9 most similar retrieved images, green for relevant images, red otherwise. Top line results with ROADMAP. Bottom line results with SmoothAP.

## 5   Conclusion

This paper introduces the ROADMAP method for gradient-based optimization of average precision. ROADMAP is based on a smooth rank approximation, leading to the $\mathcal{L}_{\text{SupAP}}$ being both accurate and robust. To overcome the lack of decomposability in AP, ROADMAP is equipped with a calibration loss $\mathcal{L}_{\text{calibr.}}$ which aims at reducing the decomposability gap. We provide theoretical guarantees as well as experiments to assess this behavior. Experiments show that ROADMAP can combine the strength of ranking methods with the simplicity of a batch strategy. Without bells and whistles, ROADMAP is able to outperform state-of-the-art performances on three datasets, and remains effective even with small batch sizes.

As any work on image retrieval, our contribution could be applied to critical applications in surveillance scenarios, *e.g.* face recognition or person re-identification. ROADMAP is neither worse nor better than previous work in this regard. Our work is also a data-driven learning method, and thus inherits the risk of perpetuating dataset biases. Future work will focus on improving fair and accurate retrieval by reducing dataset biases. We also plan to relax the need for full supervision to tackle situations more representative to in-the-wild scenarios.

**Acknowledgement**   This work was done under a grant from the the AHEAD ANR program (ANR-20-THIA-0002). It was granted access to the HPC resources of IDRIS under the allocation 2021-AD011012645 made by GENCI.

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
