# Robust and Decomposable Average Precision for Image Retrieval
## – Supplementary Material –

**Elias Ramzi**[1,2]
elias.ramzi@cnam.fr

**Nicolas Thome**[1]
nicolas.thome@cnam.fr

**Clément Rambour**[1]
clement.rambour@cnam.fr

**Nicolas Audebert**[1]
nicolas.audebert@cnam.fr

**Xavier Bitot**[2]
xavier.bitot@coexya.eu

[1]CEDRIC, Conservatoire National des Arts et Métiers, Paris, France
[2]Coexya, Paris, France

## A  ROADMAP model

### A.1  Properties of SupAP & comparison to SmoothAP

We further discuss and give additional explanations of the property of our $\mathcal{L}_{\text{SupAP}}$ loss function, and especially its comparison with respect to the SmoothAP [1] baseline.

As shown in Fig. 1.a of the main paper, and discussed in Section 3.1 ("Comparison to SmoothAP"), the smooth rank approximation in [1] has several drawbacks, that we show below:

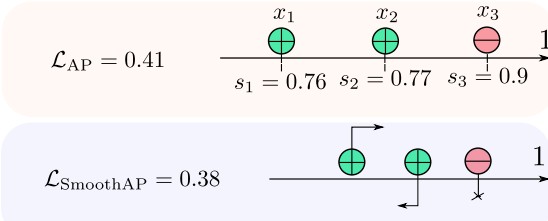

Figure 1: Limitation of the smooth rank approximation in [1]: contradictory gradient flow for the positives samples $x_1$ and $x_2$ (in green), vanishing gradient for the negative example $x_3$ (in red), and no guarantees of having an upper bound of $\mathcal{L}_{\text{AP}}$.

Specifically, we explain in more detail the following three limitations identified in the main paper for SmoothAP [1], which comes from the use of the sigmoid function to approximate the Heaviside (step) function for computing the rank:

i **Contradictory gradient flow for positives samples:** Firstly we can see on the toy dataset of Fig. 1 that the gradients of the two positive examples (in green) with SmoothAP have opposite directions. The positive with the lowest rank $x_1$ has a gradient in the good direction, since it leads to increase $x_1$'s score because the correct ordering is not reached (the negative instance

$x_3$ has a better rank). But the gradient of the positive with the highest rank $x_2$ is on the wrong direction, since it tends to decrease $x_2$'s score. This is an undesirable behaviour, which comes from the use of the sigmoid in $\mathcal{L}_{\text{SmoothAP}}$. In the example of Fig. 1, we can actually show that

$$\boxed{\frac{\partial \mathcal{L}_{\text{SmoothAP}}}{\partial s_1} = -\frac{\partial \mathcal{L}_{\text{SmoothAP}}}{\partial s_2}}$$

To see this we write :

$$\frac{\partial \mathcal{L}_{\text{SmoothAP}}}{\partial s_1} = \frac{\partial \mathcal{L}_{\text{SmoothAP}}}{\partial \text{rank}^+(x_1)} \cdot \frac{\partial \text{rank}^+(x_1)}{\partial s_1} + \frac{\partial \mathcal{L}_{\text{SmoothAP}}}{\partial \text{rank}^+(x_2)} \cdot \frac{\partial \text{rank}^+(x_2)}{\partial s_1}$$
$$+ \frac{\partial \mathcal{L}_{\text{SmoothAP}}}{\partial \text{rank}^-(x_1)} \cdot \frac{\partial \text{rank}^-(x_1)}{\partial s_1} + \frac{\partial \mathcal{L}_{\text{SmoothAP}}}{\partial \text{rank}^-(x_2)} \cdot \frac{\partial \text{rank}^-(x_2)}{\partial s_1}$$

Because $\text{rank}^-(x_2) = \sigma(\frac{s_3-s_2}{\tau})$, we have $\frac{\partial \text{rank}^-(x_2)}{\partial s_1} = 0$ and $\frac{\partial \text{rank}^-(x_1)}{\partial s_1} = 0$ in the example of Fig. 1, because $\text{rank}^-(x_1) = \sigma(\frac{s_3-s_1}{\tau})$ and $s_3 - s_1$ falls into the saturation regime of the sigmoid. We get a similar result for the derivative of $\mathcal{L}_{\text{SmoothAP}}$ wrt. $s_2$ :

$$\frac{\partial \mathcal{L}_{\text{SmoothAP}}}{\partial s_2} = \frac{\partial \mathcal{L}_{\text{SmoothAP}}}{\partial \text{rank}^+(x_1)} \cdot \frac{\partial \text{rank}^+(x_1)}{\partial s_2} + \frac{\partial \mathcal{L}_{\text{SmoothAP}}}{\partial \text{rank}^+(x_2)} \cdot \frac{\partial \text{rank}^+(x_2)}{\partial s_2}$$

Furthermore we have :

$$\frac{\partial \text{rank}^+(x_1)}{\partial s_1} = -\frac{\partial \text{rank}^+(x_1)}{\partial s_2}$$

Indeed $\text{rank}^+(x_1) = 1 + \sigma(\frac{s_2-s_1}{\tau})$, such that $\frac{\partial \text{rank}^+(x_1)}{\partial s_1} = -\tau \cdot \sigma(\frac{s_2-s_1}{\tau})\left(1 - \sigma(\frac{s_2-s_1}{\tau})\right)$ and $\frac{\partial \text{rank}^+(x_1)}{\partial s_2} = \tau \cdot \sigma(\frac{s_2-s_1}{\tau})\left(1 - \sigma(\frac{s_2-s_1}{\tau})\right)$. Similarly the derivatives of $\text{rank}^+(x_2)$ wrt. $s_1$ and $s_2$ also have opposite signs: $\frac{\partial \text{rank}^+(x_2)}{\partial s_1} = -\frac{\partial \text{rank}^+(x_2)}{\partial s_2}$. It concludes the proof that $\frac{\partial \mathcal{L}_{\text{SmoothAP}}}{\partial s_1} = -\frac{\partial \mathcal{L}_{\text{SmoothAP}}}{\partial s_2}$.

ii **Vanishing gradients:** Secondly, SmoothAP [1] has vanishing gradients due to its use of the sigmoid function. This is illustrated on the toy dataset in Fig. 1. The negative instance $x_3$ has a high score $s_3$, but does not receive any gradient, which does not enable it to lower its score although it would improve the overall ranking. This is because the score difference between $x_3$ and $x_2$ is large, *i.e.* $s_3 - s_2 = 0.13$. Similarly, $s_3 - s_1 = 0.14$. Consequently, both $s_3 - s_2$ and $s_3 - s_1$ fall into the saturation regime of the sigmoid, preventing to propagate any gradient (see Fig. 3c. in the main paper).

iii **Finally, $\mathcal{L}_{\text{SmoothAP}}$ is not an upper bound of $\mathcal{L}_{\text{AP}}$.** The use of the sigmoid means that both $\text{rank}^+$ and $\text{rank}^-$ can be over or under estimated. If $\text{rank}^+$ is overestimated (resp. underestimated) $\mathcal{L}_{\text{SmoothAP}}$ underestimates $\mathcal{L}_{\text{AP}}$ (resp. overestimates). And if $\text{rank}^-$ is overestimated (resp. underestimated) $\mathcal{L}_{\text{SmoothAP}}$ overestimates $\mathcal{L}_{\text{AP}}$ (resp. overestimated). Therefore, $\mathcal{L}_{\text{SmoothAP}}$ can be larger or lower than $\mathcal{L}_{\text{AP}}$ in general. In the example of Fig. 1, we show that $\mathcal{L}_{\text{SmoothAP}}$ is lower than $\mathcal{L}_{\text{AP}}$.

**We address those three issues with $\mathcal{L}_{\text{SupAP}}$:**

i **Using the the true Heaviside (step) function $\mathbf{H}^+$ for $\text{rank}^+$** allows to have the expected behaviour regarding the gradients of positives. When Changing $\mathbf{H}^+$ for $\text{rank}^+$ in Fig. 2a, we can see that we fix the problem of opposite gradients for the positive examples $x_1$ and $x_2$ - although the gradient is zero.

ii **Using $\mathbf{H}^-$ for $\text{rank}^-$ overcomes vanishing gradients**. By using $\mathbf{H}^-$ in Eq. (4) in submission, we design a linear function for positive $(s_j - s_k)$ values, where $s_j$ (resp. $s_k$) is the score of a negative (resp. positive) example - see Fig. 3b in the main paper. We can see in Fig. 2b that this change enables to have gradients in the correct directions for the two positive instances $x_1$ and $x_2$ (tending to increase their scores), and for the negative instance $x_3$ (tending to decrease its score).

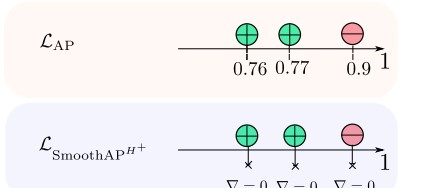 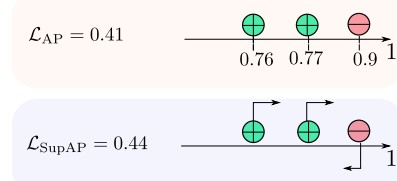

(a) When replacing $H^+$ by the Heaviside function in SmoothAP we stop the unexpected behaviour of the gradient flow. However there is still vanishing gradients.

(b) Our $\mathcal{L}_{\text{SupAP}}$ has gradients that do not stop until the correct ranking is achieved.

Figure 2: We illustrates the different steps to built $\mathcal{L}_{\text{SupAP}}$. On Fig. 2a we change $H^+$ to be the true Heaviside (step) function. On Fig. 2b we replace the sigmoid by $H^-$ defined in Eq. (4) of the main paper. Using $H^+$ and $H^-$, $\mathcal{L}_{\text{SupAP}}$ is an upper bound of $\mathcal{L}_{\text{AP}}$.

iii **$\mathcal{L}_{\text{SupAP}}$ is an upper bound of $\mathcal{L}_{\text{AP}}$.** By the proposed design of $\mathbf{H}^-$ in Eq. (4) in submission, we have $\text{rank}_s^-(k) \geq \text{rank}^-(k)$. Since we do not approximate $\text{rank}^+(k)$ by keeping the Heaviside function, it leads to $\frac{\text{rank}^+(k)}{\text{rank}^+(k)+\text{rank}_s^-(k)} \leq \frac{\text{rank}^+(k)}{\text{rank}^+(k)+\text{rank}^-(k)}$, and therefore $\mathcal{L}_{\text{SupAP}} \geq \mathcal{L}_{\text{AP}}$.

Overall, $\mathcal{L}_{\text{SupAP}}$ has all the desired properties : i) A correct gradient flow during training, ii) No vanishing gradients while the correct ranking is not reached, iii) Being an upper bound on the AP loss $\mathcal{L}_{\text{AP}}$.

## A.2 Properties of the $\mathcal{L}_{\text{calibr.}}$ loss function

We remind the reader of the definition of the decomposability gap given in Eq. (6) of the main paper.

$$DG_{\text{AP}}(\boldsymbol{\theta}) = \frac{1}{K}\sum_{b=1}^{K} \text{AP}_i^b(\boldsymbol{\theta}) - \text{AP}_i(\boldsymbol{\theta})$$

We illustrates the decomposability gap, $DG_{AP}$ with the toy dataset of Fig. 3. The decomposability gap comes from the fact that the AP is not decomposable in mini-batches as we discuss in the Sec. 3.2 of the main paper. The motivation behind $\mathcal{L}_{\text{calibr.}}$ is thus to force the scores of the different batches to aligned as illustrated in the Fig. 2b of the main paper.

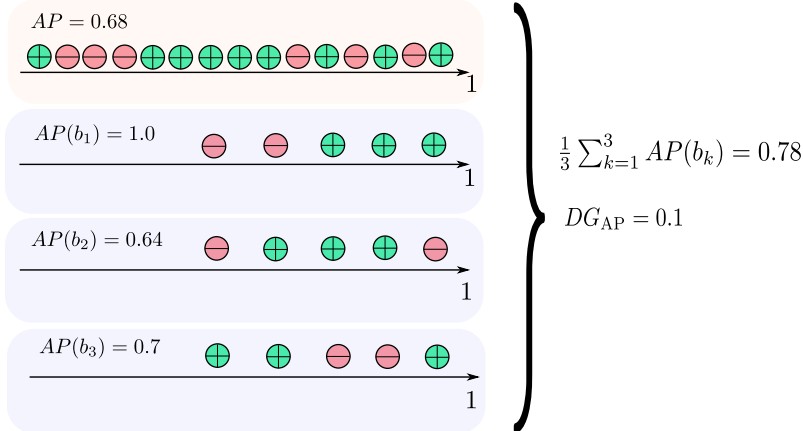

Figure 3: Illustration of the decomposability gap on a toy dataset.

**Proof of Eq. (8): Upper bound on the $DG_{\text{AP}}$ with no $\mathcal{L}_{\text{AP}}$**   We choose a setting for the proof of the upper bound similar to the one used for training, *i.e.* all the batch have the same size, and the number of positive instances per batch (*i.e.* $\mathcal{P}_i^b$) is the same.

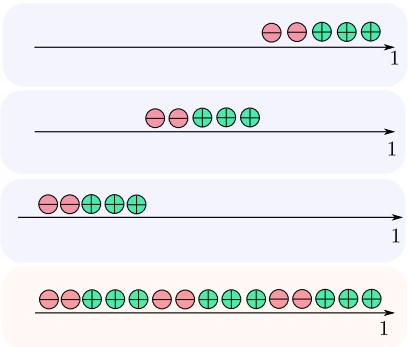

Figure 4: The worst case when computing the global AP would be that each batch is juxtaposed.

Eq. (8) from the main paper gives an upper bound for $DG_{AP}$. This upper bound is given in the worst case: when the AP has the lowest value guaranteed by the AP on each batch. We illustrate this case in Fig. 4.

In Eq. (8) from the main paper the 1 in the right hand term comes from the average of AP over all batches:

$$\frac{1}{K} \sum_{b=1}^{K} AP_i^b(\theta) = 1$$

We then justify the term in the parenthesis of Eq. (8) in the main paper, which is the lower bound of the AP. In the global ordering the positive instances are ranked after all the positive instances from previous batches giving the following $\mathrm{rank}^+$: $j + |\mathcal{P}_i^1| + \cdots + |\mathcal{P}_i^{b-1}|$, with $j$ the $\mathrm{rank}^+$ in the batch, Positive instances are also ranked after all negative instances from previous batches giving $\mathrm{rank}^-$: $|\mathcal{N}_i^1| + \cdots + |\mathcal{N}_i^{b-1}|$.

Therefore we obtain the resulting upper bound of Eq. (8) of the main paper:

$$0 \le DG_{\mathrm{AP}} \le 1 - \frac{1}{\sum_{b=1}^{K} |\mathcal{P}_i^b|} \left( \sum_{b=1}^{K} \sum_{j=1}^{B} \frac{j + |\mathcal{P}_i^1| + \cdots + |\mathcal{P}_i^{b-1}|}{j + |\mathcal{P}_i^1| + \cdots + |\mathcal{P}_i^{b-1}| + |\mathcal{N}_i^1| + \cdots + |\mathcal{N}_i^{b-1}|} \right)$$

**Proof of Eq. (9): Upper bound on the $DG_{AP}$ with $\mathcal{L}_{\mathbf{AP}}$** In the main paper we refine the upper bound on $DG_{AP}$ in Eq. (9) by adding $\mathcal{L}_{\mathrm{calibr.}}$ which calibrates the absolute scores across the mini-batches.

We now write that each positive instance that respects the constraint of $\mathcal{L}_{\mathrm{calibr.}}$ is ranked after the positive instances of previous batch that respect the constraint giving the following $\mathrm{rank}^+$: $j + G_1^+ + \cdots + G_{b-1}^+$, with $j$ the $\mathrm{rank}^+$ in the current batch. Positive instances are also ranked after the negative instances of previous batches that do not respect the constraints yielding $\mathrm{rank}^-$ : $E_1^- + \cdots + E_{b-1}^-$.

We then write that positive instances that do not respect the constraints are ranked after all positive instances from previous batches and the positive instances respecting the constraints of the current batch giving $\mathrm{rank}^+$ : $j + G_b^+ |\mathcal{P}_i^1| + \cdots + |\mathcal{P}_i^{b-1}|$. They also are ranked after all the negative instances from previous batches giving $\mathrm{rank}^-$ : $|\mathcal{N}_i^1| + \cdots + |\mathcal{N}_i^{b-1}|$.

Resulting in Eq. (9) from the main paper:

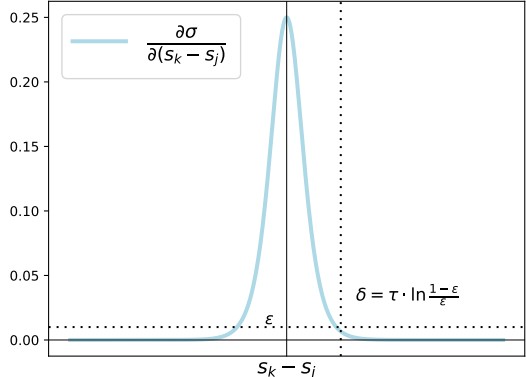

Figure 5: Gradient of the temperature scaled sigmoid ($\tau = 0.01$) *vs.* the difference of scores $s_k - s_j$ of a negative pair.

$$0 \leq DG_{\text{AP}} \leq 1 - \frac{1}{\sum_{b=1}^{K} |\mathcal{P}_i^b|} \left( \sum_{b=1}^{K} \left[ \sum_{j=1}^{G_b^+} \frac{j + G_1^+ + \cdots + G_{b-1}^+}{j + G_1^+ + \cdots + G_{b-1}^+ + E_1^- + \ldots E_{b-1}^-} + \right. \right.$$

$$\left. \left. \sum_{j=1}^{E_b^+} \frac{j + G_b^+ + |\mathcal{P}_i^1| + \cdots + |\mathcal{P}_i^{b-1}|}{j + G_b^+ + |\mathcal{P}_i^1| + \cdots + |\mathcal{P}_i^{b-1}| + |\mathcal{N}_i^1| + \cdots + |\mathcal{N}_i^{b-1}|} \right] \right)$$

### A.3 Choice of $\delta$

In the main paper we introduce $\delta$ in Eq. (4) to define $H^-$. We choose $\delta$ as the point where the gradient of the sigmoid function becomes low $< \epsilon$, and we then have $\delta = \tau \cdot \ln \frac{1-\epsilon}{\epsilon}$. This is illustrated in Fig. 5. For our experiments we use $\epsilon = 10^{-2}$ giving $\delta \simeq 0.05$.

## B Experiments

### B.1 Metrics

We detail here the performance metrics that we use to evaluate our models.

**Recall@K** The Recall@K metrics is often used in the literature. For a single query the Recall@K is 1 if a positive instance is in the K nearest neighbors, and 0 otherwise. The Recall@K is then averaged on all the queries. Researcher use different values of K for a given dataset (*e.g.* 1, 2, 4, 8 on CUB).

$$R@K = \frac{1}{M} \sum_{i=1}^{M} r(i), \quad \text{where } r(i) = \begin{cases} 1 \text{ if a positive instance has a ranking smaller than i} \\ 0 \quad \text{otherwise} \end{cases} \tag{1}$$

**mAP@R** Recently, the mAP@R has been introduced in [9]. The authors show that this metric is less noisy and better captures the performance of a model. The mAP@R is a partial AP, computed on the R first instances retrieved, with R being set to the number of positive instances wrt. a query.

mAP@R is a lower bound of the AP (mAP@R = AP when the correct ranking is achieved, *i.e.* mAP@R = AP = 1).

$$mAP@R_i = \frac{1}{R}\sum_{j=1}^{R} P(j), \quad \text{where } P(j) = \begin{cases} \text{precision at j if the jth retrieval is correct} \\ 0 \quad \text{otherwise} \end{cases} \tag{2}$$

## B.2 Detail on experimental setup

In this section, we describe the experimental setup used in the Sec. 4.1 of the main paper, and the Sec. B of the supplementary.

We use standard data augmentation strategy during training: images are resized so that their shorter side has a size of 256, we then make a random crop that has a size between 40 and 256, and aspect ratio between 3/4 and 4/3. This crop is then resized to 224x224, and flipped horizontally with a 50% chance. During evaluation, images are resized to 256 and then center cropped to 224.

We use two different strategy to sample each mini-batch. On CUB and INaturalist we choose a batch size (*e.g.* 128) and a number of samples per classes (*e.g.* 4). We then randomly sample classes (*e.g.* 32) to construct our batches. For SOP we use the hard sampling strategy from [2]. For each pair of category (*e.g.* bikes and coffee makers) we use the preceding sampling strategy. This sampling techniques is used because it yields harder and more informative batches. The intuition behind this sampling is that it will be harder to discriminate two bikes from one another, than a bike and a sofa.

We train the ResNet-50 models using Adam [7]. On CUB we train our models with a learning rate of $10^{-6}$ for 200 epochs. For SOP and INaturalist we take the same scheduling as in [1]. We set the learning rate for the backbone to $10^{-5}$ and the double for the added linear projection layer. We drop the learning rate by 70% on the epochs 30 and 70. Finally the models are trained for 100 epochs on SOP and 90 on INaturalist (as in [1]).

We train the DeiT transformers models using AdamW [8] as in [4]. On INaturalist we use the same schedule as when training ResNet-50, with a learning rate of $10^{-5}$. On SOP we train for 75 epochs with a learning rate of $10^{-5}$ which is dropped by 70% at epochs 25 and 50. Finally on CUB we train the models for about 100 epochs with a learning rate of $10^{-6}$.

## B.3 Details of the backbones used

We briefly describe the backbones used throughout out the experiments presented in the main paper and the supplementary.

**ResNet-50 [5]**   We use the well-known convolutional neural network ResNet-50. We remove the linear classification layer. We also add a linear projection layer to reduce the dimension (*e.g.* from 2048 to 512).

**DeiT [16]**   Recently transformer models have been introduced for computer vision [3, 16]. They establish new state-of-the-art performances on computer vision tasks. We use the DeiT-S from [16] which has less parameters than the ResNet-50 ($\sim$ 21 million for DeiT *vs.* 25 for ResNet-50). We use the pretrained version with distillation from [16] and its implementation in the `timm` library [17].

## B.4 ROADMAP validation

**Comparison to AP approximations**   We compare in Table 1 ROADMAP *vs.* other ranking losses on different settings : a batch size of 128 and two backbones (ResNet-50 and DeiT). We conduct this comparison on 5 runs to show the statistical improvement of our method compared to other ranking losses baselines.

We observe that our method outperforms recent ranking losses on the two backbones and the three datasets. On SOP and CUB, ROADMAP has a high increase for the mAP@R, of +1pt on CUB and +2pt on SOP. The performance improvement is greater on the large scale dataset INaturalist with $\sim$+3.5pt with a ResNet-50 backbone and $\sim$+2pt with a DeiT backbone of mAP@R. This trend is the same as in the comparison of the main paper (Table 1).

Table 1: Comparison between ROADMAP and state-of-the-art AP ranking based losses on three image retrieval datasets. *Bck* in the first column stands for bakcbone. Models are trained with a batch size of 128.

| Bck | Method | CUB R@1 | CUB mAP@R | SOP R@1 | SOP mAP@R | INaturalist R@1 | INaturalist mAP@R |
|---|---|---|---|---|---|---|---|
| ResNet-50 | FastAP [2] | 61.28±0.37 | 24.11±0.16 | 78.97±0.05 | 52.23±0.09 | 57.23±0.05 | 22.17±0.05 |
| | SoftBinAP [14] | 61.70±0.10 | 24.29±0.16 | 80.30±0.21 | 53.69±0.27 | 60.88±0.06 | 23.22±0.05 |
| | BlackBoxAP [15] | 61.96±0.28 | 23.83±0.14 | 80.97±0.07 | 54.49±0.15 | 59.53±0.12 | 19.62±0.02 |
| | SmoothAP [1] | 62.45±0.48 | 24.32±0.1 | 81.13±0.05 | 54.74±0.16 | 64.48±0.05 | 24.33±0.07 |
| | ROADMAP (ours) | **64.05**±0.51 | **25.27**±0.12 | **82.20**± 0.09 | **56.64**±0.09 | **68.15**±0.10 | **27.01**±0.10 |
| DeiT | FastAP [2] | 73.42±0.22 | 31.96±0.06 | 82.92±0.07 | 59.06±0.03 | 62.18±0.07 | 25.48±0.10 |
| | SoftBinAP [14] | 74.84±0.11 | 33.57±0.08 | 84.09±0.05 | 60.53±0.07 | 65.97±0.13 | 27.57±0.09 |
| | BlackBoxAP [15] | 75.45±0.22 | 33.97±0.10 | 84.07±0.09 | 60.20±0.05 | 70.29±0.10 | 29.44±0.06 |
| | SmoothAP [1] | 76.02±0.14 | 34.69±0.08 | 84.28±0.06 | 60.49±0.17 | 69.80±0.08 | 29.56±0.04 |
| | ROADMAP (ours) | **77.14**±0.12 | **36.30**±0.08 | **85.44**± 0.06 | **62.73**±0.06 | **72.81**±0.11 | **31.31**±0.10 |

Table 2: Ablation study for the impact of our two contribution *vs.* the SmoothAP baseline for the three datasets and different batch sizes, with a ResNet-50 backbone [5]

| BS | Method | $H^-$ | $\mathcal{L}_{calibr.}$ | CUB R@1 | CUB mAP@R | SOP R@1 | SOP mAP@R | INaturalist R@1 | INaturalist mAP@R |
|---|---|---|---|---|---|---|---|---|---|
| 32 | SmoothAP | ✗ | ✗ | 61.84 | 23.76 | 79.96 | 53.21 | 53.25 | 16.4 |
| | SupAP | ✓ | ✗ | 62.58 | 24.12 | 80.51 | 53.85 | 55.01 | 17.13 |
| | ROADMAP | ✓ | ✓ | **63.69** | **24.97** | **80.74** | **54.68** | **56.43** | **20.43** |
| 128 | SmoothAP | ✗ | ✗ | 62.81 | 24.44 | 81.19 | 54.96 | 64.53 | 24.26 |
| | SupAP | ✓ | ✗ | 63.18 | 24.9 | 81.72 | 55.65 | 65.79 | 24.77 |
| | ROADMAP | ✓ | ✓ | **64.18** | **25.38** | **82.18** | **56.64** | **68.28** | **27.13** |
| 224 | SmoothAP | ✗ | ✗ | 62.93 | 24.69 | 81.2 | 54.73 | 66.62 | 26.08 |
| | SupAP | ✓ | ✗ | 64.08 | 25.13 | 81.88 | 55.75 | 67.43 | 26.32 |
| | ROADMAP | ✓ | ✓ | **64.65** | **25.51** | **82.3** | **56.55** | **69.28** | **27.74** |
| 384 | SmoothAP | ✗ | ✗ | 63.69 | 24.89 | 81.45 | 55.1 | 67.39 | 26.77 |
| | SupAP | ✓ | ✗ | 64.64 | 25.27 | 81.94 | 55.78 | 68.37 | 27.24 |
| | ROADMAP | ✓ | ✓ | **64.69** | **25.36** | **82.31** | **56.47** | **69.19** | **27.85** |

We perform a paired student t-test to further asses the statistical significance of the performance boost obtained with ROADMAP. We compute the p-values for both the R@1 and mAP@R: it turns out that the p-values are never larger than $0.001$, meaning that the gain is statistically significant (with a risk less than 0.1%).

**Ablation studies**    In Table 2 we extend the ablation studies of the main paper (Table 2 of main paper) to other settings, including more batch sizes (32, 128, 224, 384) and two backbones (ResNet-50 and DeiT). On all settings $\mathcal{L}_{SupAP}$ outperforms the $\mathcal{L}_{SmoothAP}$ baseline by almost ∼+0.5pt consistently, and almost +1pt on every setting for INaturalist. When we add $\mathcal{L}_{calibr.}$ the gain is further increased. As noticed in Table 2 (main paper) the gain when adding $\mathcal{L}_{calibr.}$ is particularly noticeable on the large scale dataset INaturalist with boost in performances that can be up to +3.3pt of mAP@R for the ResNet-50 with a batch size 32.

In Table 3 we extend ablation studies with a transformer backbone (DeiT). We observe the same trend as in Table 2. $\mathcal{L}_{SupAP}$ is consistently better than the $\mathcal{L}_{SmoothAP}$ baseline, with gain up to more than 1pt (*e.g.* on batch size 128 on INaturalist). $\mathcal{L}_{calibr.}$ further lifts the performances on the three datasets and all batch sizes.

**Comparison to state of the art method**    We show in Table 4 the impact of increasing the embedding dimension when using ResNet-50. All metrics improve on the three datasets when the

Table 3: Ablation study for the impact of our two contribution *vs.* the SmoothAP baseline for the three datasets and different batch sizes, with a DeiT backbone [16]

| BS | Method | $H^-$ | $\mathcal{L}_{calibr.}$ | CUB | | SOP | | INaturalist | |
|---|---|---|---|---|---|---|---|---|---|
| | | | | R@1 | mAP@R | R@1 | mAP@R | R@1 | mAP@R |
| | SmoothAP | ✗ | ✗ | 76.2 | 34.7 | 84.16 | 60.18 | 69.83 | 29.49 |
| 128 | SupAP | ✓ | ✗ | 76.33 | 34.91 | 84.74 | 61.29 | 71.12 | 30.5 |
| | ROADMAP | ✓ | ✓ | **77.09** | **35.76** | **85.44** | **62.57** | **72.82** | **31.36** |
| | SmoothAP | ✗ | ✗ | 76.38 | 35.33 | 84.3 | 60.49 | 70.55 | 30.25 |
| 224 | SupAP | ✓ | ✗ | 76.47 | 35.67 | 84.77 | 61.38 | 71.9 | 31.31 |
| | ROADMAP | ✓ | ✓ | **77.14** | **36.18** | **85.56** | **62.75** | **73.64** | **31.82** |
| | SmoothAP | ✗ | ✗ | 76.72 | 35.86 | 84.66 | 61.26 | 71.09 | 30.89 |
| 384 | SupAP | ✓ | ✗ | 77.13 | 36.17 | 85.01 | 61.76 | 72.55 | 31.89 |
| | ROADMAP | ✓ | ✓ | **77.38** | **36.23** | **85.35** | **62.29** | **73.64** | **32.12** |

Table 4: Difference in performance when using an embedding size of 512 *vs.* 2048 with a ResNet-50 backbone, on the three datasets. Performances are obtained with the same setup as described in the Sec. 4.2 of the main paper.

| Method | dim | CUB | | SOP | | INaturalist | |
|---|---|---|---|---|---|---|---|
| | | R@1 | mAP@R | R@1 | mAP@R | R@1 | mAP@R |
| ROADMAP (ours) | 512 | 68.5 | 27.97 | 83.19 | 58.05 | 69.19 | 27.85 |
| ROADMAP (ours) | 2048 | **69.87** | **28.8** | **83.77** | **59.38** | **69.62** | **27.87** |

embedding dimension increases. We observe a gain particularly important on CUB and SOP with ∼+1pt in R@1 and mAP@R.

Choosing an embedding size of 2048 further boost the performances of ROADMAP, yielding competitive performances on CUB and state-of-the-art performances for SOP and INaturalist.

**Preliminary results on Landmarks retrieval** We show in Table 5 preliminary experiments to evaluate ROADMAP on $\mathcal{R}$Oxford and $\mathcal{R}$Paris [13], by training our model on the SfM-120k dataset and using the standard GitHub code for evaluation[1].

We can see that ROADMAP is significantly better than [4] with the DeiT-S [16] on $\mathcal{R}$Oxford and $\mathcal{R}$Paris medium protocol, and has similar performances for $\mathcal{R}$Paris hard protocol. This highlights the relevance of using ROADMAP instead of the contrastive loss used in [4].

Table 5: Comparison of ROADMAP vs IRT [4] on $\mathcal{R}$Oxford and $\mathcal{R}$Paris [13]. Models are DeiT-S [16], ROADMAP is trained with a batch size of 128.

| Method | $\mathcal{R}$Oxford | | $\mathcal{R}$Paris | |
|---|---|---|---|---|
| | Medium | Hard | Medium | Hard |
| IRT [4] | 34.5 | 15.8 | 65.8 | 42.0 |
| ROADMAP (ours) | **38.9** | **20.7** | **67.5** | **42.3** |

---

[1]`https://github.com/filipradenovic/cnnimageretrieval-pytorch`

## B.5  Model analysis

**Hyperparameters**   In Fig. 6 we show the impact of the hyperparameters of $\mathcal{L}_{\text{SupAP}}$. We plot the mAP@R *vs.* $\tau$ in Fig. 6a and mAP@R *vs.* $\rho$ in Fig. 6b. The experiments are conducted on SOP with a batch size of 128.

We observe on Fig. 6a that $\mathcal{L}_{\text{SupAP}}$ is stable with small values of $\tau$, *i.e.* in the range [0.001, 0.05]. As a reminder we use the default value $\tau = 0.01$ in all our results, as it was the suggested value from the SmoothAP paper [1].

We conduct a study of the impact of $\rho$ in Fig. 6b. We find that $\mathcal{L}_{\text{SupAP}}$ is very stable wrt. this hyperparameter. Performances are improving with a greater value of $\rho$ before dropping after $10^4$. The trend follows what was observed in the Fig. 4b of the main paper, although this time using a value if $\rho = 10^4$ yields better performances. Using cross-validation to choose an optimal value for $\rho$ may lead to even better performances for $\mathcal{L}_{\text{SupAP}}$.

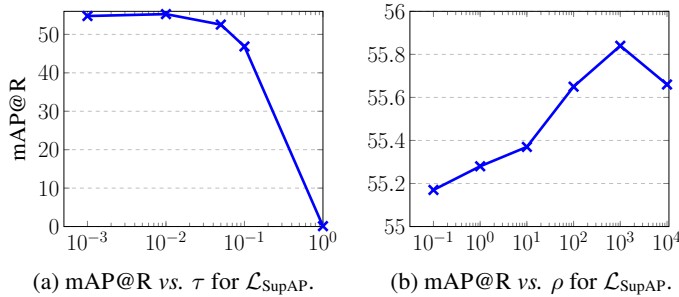

(a) mAP@R *vs.* $\tau$ for $\mathcal{L}_{\text{SupAP}}$.     (b) mAP@R *vs.* $\rho$ for $\mathcal{L}_{\text{SupAP}}$.

Figure 6: Analysis of $\mathcal{L}_{\text{SupAP}}$ hyperparameters on SOP (batch size 128).

**Decomposability gap**   In Table 6 we measure the relative decrease of the decomposability gap $DG_{AP}$ on SOP and CUB test sets. On both datasets we can see that $\mathcal{L}_{\text{calibr.}}$ decreases the decomposability gap.

Table 6: Relative decrease of the decomposability gap when adding $\mathcal{L}_{\text{calibr.}}$ to $\mathcal{L}_{\text{SupAP}}$ (ROADMAP).

| Dataset | decrease of $DG_{AP}$ |
|---------|------------------------|
| CUB     | 3.7%                   |
| SOP     | 5.4%                   |

## B.6  Source code

We describe in this section the software used for our work, and discuss the computation costs associated with training models presented in this paper.

**Librairies**   We use several Python libraries often used in image retrieval.

We use `PyTorch` [12] as a general framework to implement our neural networks, losses and training loops. We use several utilities from `PyTorch Metric Learing` [10], an open-source Python library focused on helping researcher working on image retrieval and metric learning. We use `Faiss` [6] to compute metrics (*i.e.* to perform nearest neighbours search), which is a Python library often used in image retrieval to compute the rankings or the similarity matrix. To load and use the transformer models we use `timm` [17], a library implementing recent computer vision models, with pretrained weights for most of them. To handle all our config files, we use `Hydra` [18], this library makes it possible to combine the use of Yaml configuration files and overriding them using the command line.

We use the publicly available implementation of SoftBinAP[2] [14] which is under a BSD-3 license. The original codes of SmoothAP[3] [1], BlackBox[4] [11, 15] are under an MIT license. For FastAP [2] we use the implementation from [10] (MIT license), the original implementation of FastAP[5] is also under an MIT license.

**Compute costs**  We use mixed-precision learning offered within PyTorch [12]. The time and memory consumption are reduced by a factor between $2$ and $3/2$ with no notable difference in performances. We could train all models on 16GiB GPUs, except for models trained with a batch size of 384 which requires a 32GiB GPU.

**CUB** Models take between 30 minutes and 1 hour to train on a Nvidia Quadro RTX 5000 with 16GiB.

**SOP** Models take between 4 and 8 hours to train on a Nvidia Quadro RTX 5000 with 16GiB.

**INaturalist** To train models on INaturalist we were granted access to the IDRIS HPC cluster with Tesla V-100 GPUs (of 16GiB or 32GiB). Models train for approximately 20 hours.

We could not train models with mixed-precision when using BlackBox [15]. Models trained with it took longer to train (*e.g.* 30 hours on INaturalist) and are more demanding on memory (almost 16GiB with a batch size of 128 while models trained with other loss functions required less than 10Gib).

## C  Qualitative results

**CUB**  As a qualitative assessment, we show in Fig. 7 some results of ROADMAP on CUB. We show the queries (in purple) and the 10 most similar retrieved images, with relevant instances in green and irrelevant instances in red.

**SOP**  In Fig. 8 we perform the same assessment for SOP. In SOP there are fewer relevant instances per query (in average 5). So even for queries that retrieved all the relevant instances, there will be negative instances that have high ranks (in Fig. 8 ranks that are lower than 10).

**INaturalist**  Finally we show on Fig. 9 some examples of queries and the 10 most similar instances for a model trained with ROADMAP on INaturalist.

---

[2]`https://github.com/naver/deep-image-retrieval`
[3]`https://github.com/Andrew-Brown1/Smooth_AP`
[4]`https://github.com/martius-lab/blackbox-backprop`
[5]`https://github.com/kunhe/FastAP-metric-learning`

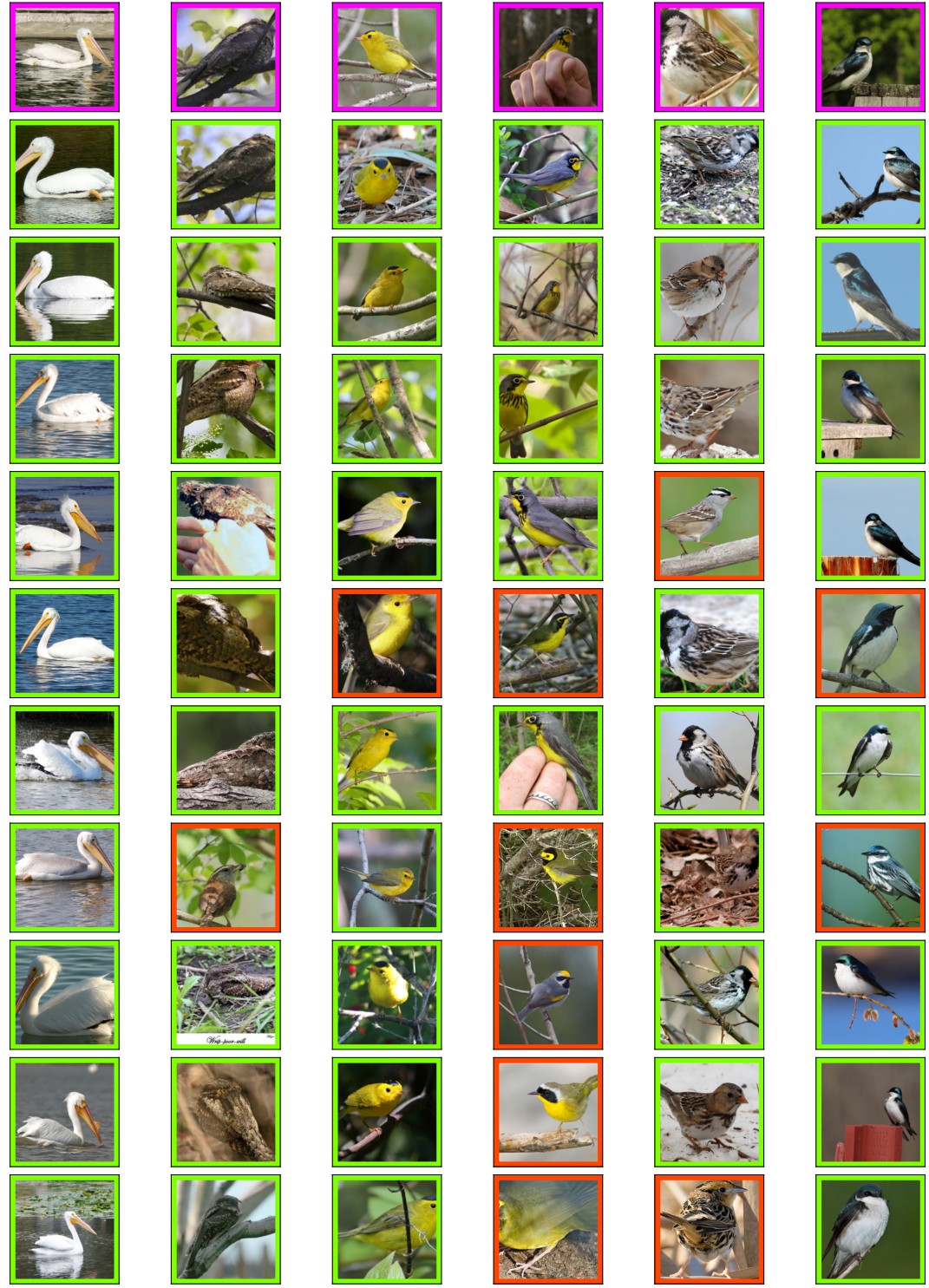

Figure 7: Qualitative results on CUB: a query (purple) with the 10 most similar instances. Relevant (resp. irrelevant) instances are in green (resp. red).

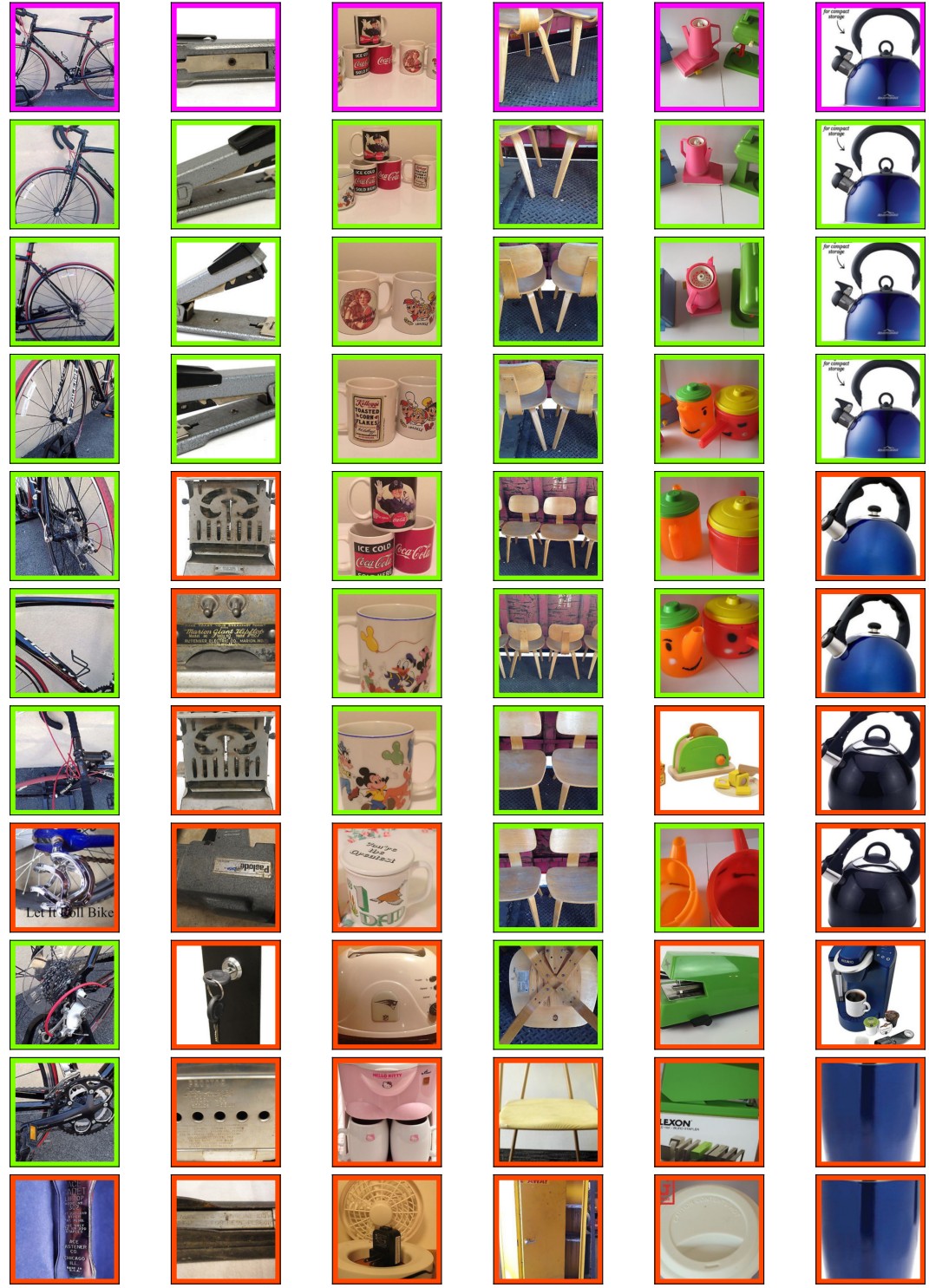

Figure 8: Qualitative results on SOP: a query (purple) with the 10 most similar instances. Relevant (resp. irrelevant) instances are in green (resp. red).

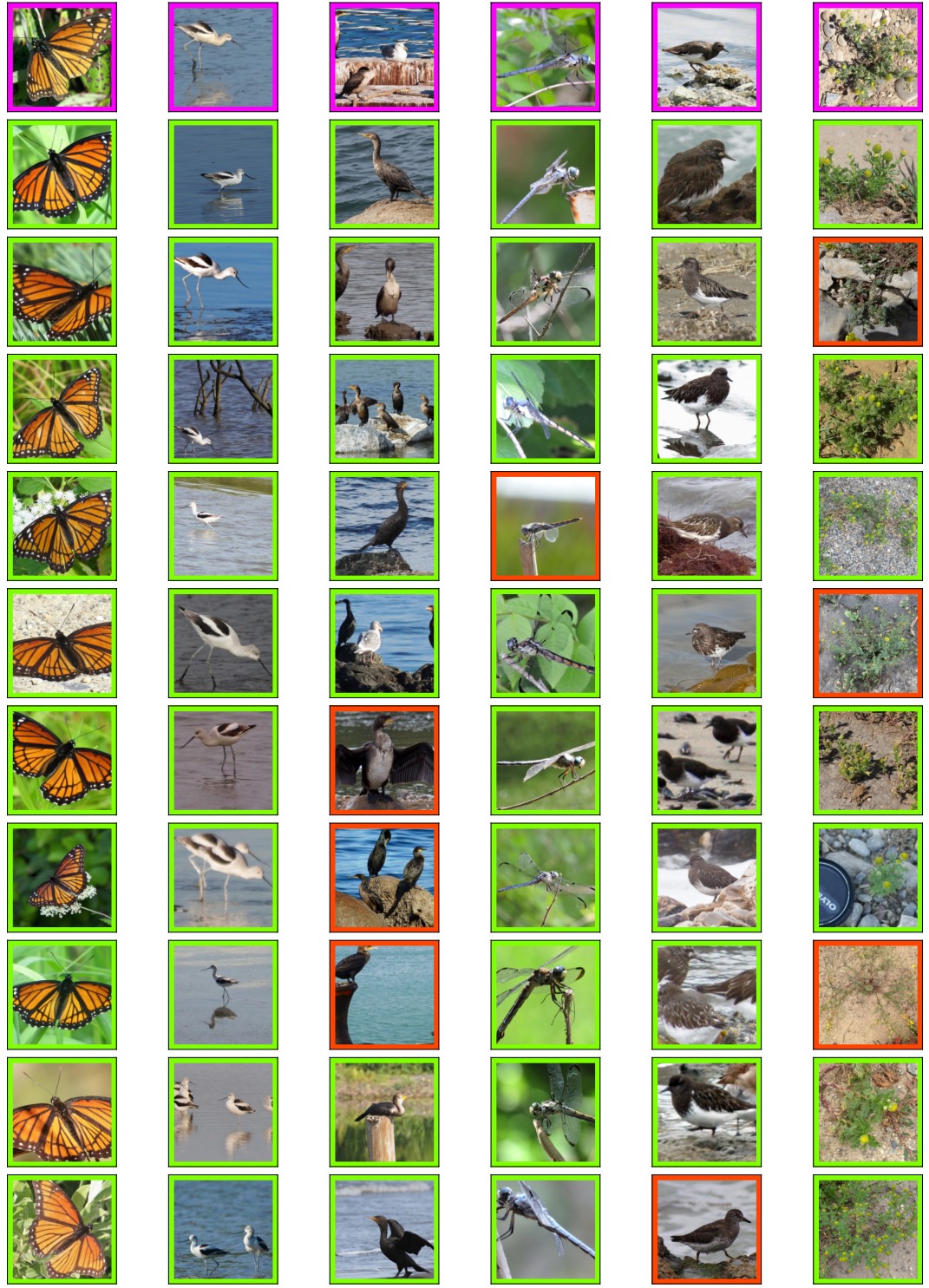

Figure 9: Qualitative results on INaturalist: a query (purple) with the 10 most similar instances. Relevant (resp. irrelevant) instances are in green (resp. red).