# OpenReview forum: "Robust and Decomposable Average Precision for Image Retrieval"
_NeurIPS.cc/2021/Conference — NeurIPS 2021 Poster_

### Official Review · Reviewer_ea3h · 2021-07-15

**Rating:** 7
**Confidence:** 3

**Summary:**

The paper proposes a new loss for image retrieval based on average precision.

Importantly, this paper proposes to address the problem in using AP using batches, which is not equivalent to the true AP computed in the whole dataset (which they refer to as 'non-decomposability'). It also proposes an improvement over other soft-binning approaches in which the version described in the paper is a proper upper-bound on the AP.

The paper lists as contributions:
1) a new surrogate loss for AP-loss that works by introducing a smooth approximation of the rank function with a different behavior for positive and negative examples,
2) improving the 'non-decomposability' problem associated with other AP-loss approximations,
3) experimental validation on CUB, SOP, and iNaturalist, showing state-of-the-art results.

**Limitations And Societal Impact:**

Yes, the authors adequately addressed the limitations and potential negative societal impact of their work.

**Main Review:**

# Strengths

The paper reads very well and has no typos. The motivations (the shortcomings of the current AP-l approximations) are well described. The approach is described well and is easy to understand. The authors report state-of-the-art results on the three datasets considered.

# Weaknesses

My main concern with the contents of the paper during the review has been the exposition of the calibration loss used to address the 'non-decomposability' problem. IIUC, the size of the non-decomposability gap is both dataset and batch-size dependent and hence proper values for \alpha \beta would also always be dataset-dependent. The paper is lacking some more information about how the values for those hyper-parameters have been selected in L.181, since, albeit referring the reader to section 4.3, section 4.3 only shows the profile of those parameters for iNaturalist. In addition, given the significant number of extra hyperparameters (\sigma, \alpha, \beta, \lambda, \tau, \rho), it would have been better if the paper evaluated the method in more datasets, e.g., RParis and ROxford, which are more standardized datasets for image retrieval. This would enable a better comparison against other works, including the AP-loss of Revaud et al, which is cited in the paper but not compared against in Table 3.



**Time Spent Reviewing:**

10

---

> ### Author Response · Authors · 2021-08-10
> **Author response to Reviewer 4**
>
> Thank you for the constructive and detailed feedback. We will address your detailed remarks in a point-by-point response below.
>
> **“My main concern with the contents of the paper during the review has been the exposition of the calibration loss used to address the 'non-decomposability' problem. IIUC, the size of the non-decomposability gap is both dataset and batch-size dependent and hence proper values for \alpha \beta would also always be dataset-dependent. The paper is lacking some more information about how the values for those hyper-parameters have been selected in L.181, since, albeit referring the reader to section 4.3, section 4.3 only shows the profile of those parameters for iNaturalist.”**
> *The calibration loss $L_{\text{Calibr.}}$  in Eq (7) depends on hyperparameters $\alpha$ and $\beta$.  $G_b^-$, $E_b^-$, $G_b^+$ and $E_b^+$ in lines 167-168 of the main paper are indeed impacted by $\beta$, $\alpha$, and the batch size. Larger batch sizes naturally lead to a smaller decomposability gap: we experimentally observe that the number of fulfilled constraints ($G_b^-$, $G_b^+$) increases with the batch size.*
>
> *Regarding $\alpha$ and $\beta$, there is a tradeoff between too small ($\alpha-\beta$) values, which do not strongly support decomposability, and too large ($\alpha-\beta$) values, which makes the training loss move away from the global AP objective. This tradeoff can be seen in Figure 4c: for $\alpha-\beta=0.6$ ROADMAP has performances (26.47 mAP@R) on par with $L_{\text{SupAP}}$ (26.32 mAP@R), and for $\alpha-\beta$ in [0.7,0.9] ROADMAP performances drop.
> As stated in the ROADMAP setting (lines 181-182 insubmission), we use a default value for all hyperparameters in our experiments ($\rho=100$, $\tau=0.01$, $\delta=0.05$, $\lambda=0.5$, $\alpha=0.9$ and $\beta=0.6$), without any attempt at finely optimizing them for a given dataset. As shown in section 4.1 and section 4.2, ROADMAP achieves very good performance with this hyperparameter setting on various contexts and datasets. In section 4.3 of the main paper, we study the impact of ($\alpha-\beta$) on INaturalist: we show in Figure 4c that the performance improvement is stable for a wide range of values ([0.1,0.6]). We show below (Table 2.) additional results obtained on SOP and CUB, for which the same trend is observed overall. On SOP, ROADMAP gets ~1pt improvement over $L_{\text{SupAP}}$ (55.6 mAP@R) in the range [0.1,0.4].  On CUB, the range for which the improvement is stable is again wide, i.e. [0.1,0.6]. We will gladly add these results in supplementary if accepted.
> Designing a more accurate setting of $\alpha$ and $\beta$ depending on dataset statistics and the batch size is an interesting perspective for future works.*
>
> Table 2. Impact of $\alpha-\beta$ on the mAP@R. Models are ResNet-50, trained with a batch size 128.
>
> |$\alpha-\beta$| 0.1 | 0.2 | 0.3 | 0.4 | 0.5 | 0.6 | 0.7 | 0.8 | 0.9 |
> |:------------:|:-----:|:-----:|:-----:|:-----:|:-----:|:-----:|:-----:|:-----:|:-----:|
> |    CUB     |24.98|25.24|25.38|25.39|25.37|25.23|23.93|20.98|18.19|
> |    SOP     |56.64|56.86|56.64|56.18|55.38|54.25|51.32|42.80|31.19|
>
> **“Given the significant number of extra hyperparameters [...],  it would have been better if the paper evaluated the method in more datasets, e.g., RParis and ROxford, which are more standardized datasets for image retrieval. This would enable a better comparison against other works, including the AP-loss of Revaud et al, which is cited in the paper but not compared against in Table 3.”**
> *We do compare ROADMAP with respect to the loss of Revaud et al. [27] in our submission, which is denoted as SoftBin in our paper, see Table 1 and Table 3. Since [27] did not perform experiments on SOP, CUB and INaturalist, reported results in our submission have been obtained by running the SoftBin loss from the authors’ GitHub code (<https://github.com/naver/deep-image-retrieval/blob/master/dirtorch/loss.py>).
> Since the submission deadline, we have run experiments on another dataset, InShop [a], on which we observed the same trends as in the 3 datasets of the main paper. Using  the same experimental setup and only changing the training loss ROADMAP obtained 91.3 vs 90.1 in R@1 for SmoothAP. Again in a similar setup to the one used in [33], ROADMAP outperforms the state-of-the-art method ProxyNCA++ (91.3 vs 90.9  R@1).
> To fulfil R4’s request, we have run preliminary experiments to evaluate ROADMAP on $\mathcal{R}$Oxford and $\mathcal{R}$Paris, by training our model on the SfM-120k dataset and using the standard GitHub code for evaluation (<https://github.com/filipradenovic/cnnimageretrieval-pytorch>). With a similar transformer backbone as in [7], we obtained the following improvements shown below:*
>
> Table 2. Comparison of ROADMAP vs IRL [7] on $\mathcal{R}$Oxford and $\mathcal{R}$Paris. Models are DeiT-S [34], ROADMAP is trained with a batch size of 128.
>
> |          | $\mathcal{R}$Ox M   | $\mathcal{R}$Ox H   | $\mathcal{R}$Par M   | $\mathcal{R}$Par H   |
> |:---------|:------:|:------:|:------:|:------:|
> | IRT [7] | 34.5 | 15.8 | 65.8 | 42.0 |
> | ROADMAP | **38.9** | **20.7** | **67.5** | **42.3** |
>
> *We can see that ROADMAP is significantly better than [7] on $\mathcal{R}$Oxford and $\mathcal{R}$Paris medium protocol, and has similar performances for RParis hard protocol. This highlights the relevance of using ROADMAP instead of the contrastive loss used in [7].*
>
> *[a] Liu, Ziwei, et al. "Deepfashion: Powering robust clothes recognition and retrieval with rich annotations." Proceedings of the IEEE conference on computer vision and pattern recognition. 2016.*

---

> > ### Comment · Reviewer_ea3h · 2021-09-10
> > **Reviewer response**
> >
> > I've read the fellow reviewers' reviews, their respective responses, and the responses to my own concerns that I had raised during my review. The authors have adequately addressed my concerns, and have included extra results on ROxford and RParis (which were also of interest for other reviewers). I am therefore keeping my initial positive rating about the work.

---

### Official Review · Reviewer_9iMr · 2021-07-15

**Rating:** 6
**Confidence:** 4

**Summary:**

This work tackles the problem of large-scale fine-grained image retrieval by proposing ROADMAP, short for ROBust And DecoMposable Average Precision which is a loss function that consists of two components
1) A smooth approximation to the non-differentiable average precision (AP) loss which provides an upper bound and better gradient flow compared to existing variants of the loss.
2) A calibration component which tackles the problem of mismatch between in-batch and true AP values commonly seen in previous methods.

The authors provided thorough theoretical proofs to support their claims on the proposed loss. Experimental results show consistent and significant improvements over existing methods in various categories, and gives state-of-the-art (SotA) performance among AP-loss based methods across three large-scale image (object) retrieval benchmarks.


**Limitations And Societal Impact:**

The authors have addressed the limitations and potential negative societal impact of this work.

**Main Review:**

## Originality
The idea of smoothening the indicator function in the AP-loss is a well-studied one as pointed out by the authors, with recent works showing success [2, 27]. The proposed smooth AP loss is very similar to the one in [2]. However, I believe the authors have given enough justification theoretically to support their differences with existing AP losses, and the analysis on the upper bounds and gradient flow is very convincing.

The idea of bridging the decomposability gap is novel, as to my best knowledge previous works tackle this with a brute-force manner, which results in either requiring huge amounts of memory or much slower training due to the need for accumulating gradients over multiple batches. I believe the contribution of this novelty is the main selling point of this work.

## Quality
This work is technically very polished, as mentioned earlier it provides highly thoughtful theoretical analysis to support the main ideas brought forward in the work. With reference to the smooth AP loss in **Originality** above, even though the improvement in this loss is incremental, the mathematical working, *e.g.* derivations of gradients in Supp.  helps a lot in the understanding of the motivation behind such proposals. One nitpick I have though: The illustrations of Fig. 1, Supp Fig. 2 & 4 are a nice addition but could be a bit not straightforward to understand. For example in Figure 1(a), why is the negative example at 0.9 and the positives at 0.76 & 0.77? Are those the $\ell_2$ distances or cosine similarities with the query? I suppose it would be the cosine similarities as the gradient direction suggests the positive (resp. negative) samples to increase (resp. decrease)? Anyway, this could cause some confusion so maybe having a small section either in the caption or the main text to explain what each component of the illustration represents, or directly annotating over the figures would help clarify this better. However, as I mentioned earlier the mathematical working more than covers for this. Especially for the upper bound of the decomposability gap in Eqs (8) & (9) which provides solid support to the addition of the proposed calibration loss term.

## Clarity
This paper is very well-written and organized, especially with the theoretical part. It also includes adequate illustrations to visualize the ideas and experimental results.

## Significance
I do not have much doubt about the significance of ROADMAP for the image retrieval community, as reflected by the SotA performance across all three datasets as presented in Tables 1 & 3. Like I mentioned earlier, the proposed variant of smooth AP loss SupAP seems to be a much more incremental step compared to the decomposability GAP loss term, as is evident in the ablation study in Table 2 the effect of adding $\mathcal{L}_\text{calibr.}$ seems to be much more pronounced than SupAP *v.s.* Smooth AP. This is especially true on INaturallist, which is by far the largest and most fine-grained (8000+ classes v.s. 200 & 12 in the other two) dataset amongst the 3 presented, which is probably the case that $\mathcal{L}_\text{calibr.}$ tackles the best due to the extra noise within batches, hence increasing $DG_\text{AP}$. One critique would be that the authors could expand on this even more, *e.g.* by gradually reducing the number of classes, do we see $\mathcal{L}_\text{calibr.}$ bring less performance boost compared to without it? Also, in the ablation study, it would also be great to see how adding $\mathcal{L}_\text{calibr.}$ alone compares with SmoothAP [2] for completeness of the analysis.

However, overall I’m still convinced by the results and adding to the thorough theoretical contributions of this work, it will definitely be a solid reference point for future works.



**Time Spent Reviewing:**

7

---

> ### Author Response · Authors · 2021-08-10
> **Author response to Reviewer 3**
>
> We thank you for your meaningful and valuable comments and the interests you took in our work. We are happy to address your remarks and questions in a point-by-point response below.
>
> **“The illustrations of Fig. 1, Supp Fig. 2 & 4 are a nice addition but could be a bit not straightforward to understand. For example in Figure 1(a), why is the negative example at 0.9 and the positives at 0.76 & 0.77? Are those the  ℓ2 distances or cosine similarities with the query?“**
> *Thank you for this remark. The values in Fig 1 indeed correspond to the cosine similarities between the examples and the query, as defined in line 94 in submission. We will clarify this  point in the final version of the paper if accepted.*
>
> **“This is especially true on INaturallist, which is by far the largest and most fine-grained (8000+ classes v.s. 200 & 12 in the other two) dataset amongst the 3 presented, which is probably the case that L_calibr tackles the best due to the extra noise within batches, hence increasing DG_AP. One critique would be that the authors could expand on this even more, e.g. by gradually reducing the number of classes, do we see Lcalibr. bring less performance boost compared to without it?”**
> *Thank you for this suggestion. Please note that although INaturalist is the most large scale dataset in terms of samples (461939 images vs. 120 053 and 11788 for the others), SOP is the most fine-grained dataset, with 22634 classes (vs. 8142 for INaturalist and 200 for CUB), each class corresponding to a specific Ebay object (eg. bike_1 or sofa_3). Each of the 22634 classes belongs to one of 12 “meta”-categories that represent a higher semantic scale (eg. bike or coffee maker).
> To us, the most important point to assess non-decomposability is the ratio between the batch size and the whole dataset size. This can be shown in Figure 5: the relative increase in mAP@R between small batch size (64) and large batch size (384) is significantly more pronounced on INaturalist (16.3-3.2=13.1pt) than on smaller scale datasets, eg. 2.4-1.2=1.2pt on SOP and 2.6-2=0.6pt on CUB.*
>
>
> **“Also, in the ablation study, it would also be great to see how adding Lcalibr. alone compares with SmoothAP [2] for completeness of the analysis.”**
> *To fulfill this request, we report below the performances of $L_{\text{Calibr.}}$ without using $L_{\text{SupAP}}$ (i.e. on top of $L_{\text{SmoothAP}}$ ), that we denote as SmoothCal. The main outcome of this experiment is that $L_{\text{Calibr.}}$ and $L_{\text{SupAP}}$ are important in different regimes.
> Indeed, $L_{\text{SmoothCal}}$  enables large gains when the  (batch size) / (dataset size) ratio is small. For example, on INaturalist with batch size 64, $L_{\text{SmoothAP}}$  is 59.8 in R@1 vs 64.3 R@1 for $L_{\text{SmoothCal}}$ loss, which is close to the 64.5 R@1 of ROADMAP and superior to 61.2 R@1 for $L_{\text{SupAP}}$. On the same batch size, the trend is different on SOP, where $L_{\text{SmoothCal}}$ reaches similar scores (81.5 R@1) compared to $L_{\text{SupAP}}$ (81.4 R@1), and using ROADMAP further brings an important gain (82 R@1).
> For larger (batch size) / (dataset size) ratios, $L_{\text{SupAP}}$ can be equivalent and sometimes superior to $L_{\text{SmoothCal}}$, e.g. $L_{\text{SmoothCal}}$ 81.5 R@1 vs $L_{\text{SupAP}}$ 81.9 R@1 on SOP (batch size 384) or $L_{\text{SmoothCal}}$ 64.0 R@1 vs $L_{\text{SupAP}}$ 64.1 R@1 on CUB (batch size 224), with ROADMAP again further improving performances (64.7 R@1).
> We will be glad to add these results in the final paper if accepted.*
>
> Table 1: Ablation studies for the impact of our two contributions and the $L_{\text{SmoothAP}}$  baseline on the three datasets. Models are ResNet-50 trained with different batch sizes.
>
> | BS  | Loss      | CUB  | SOP  | INat  |
> |:-----:|:-----------|:------:|:------:|:-------:|
> | 64  | SmoothAP  | 62.1 | 81.0 |  59.8 |
> |     | SupAP     | 62.9 | 81.4 |  61.2 |
> |     | SmoothCal | 63.7 | 81.5 |  64.3 |
> |     | ROADMAP   | **63.8** | **82.0** |  **64.5** |
> | 224 | SmoothAP  | 62.9 | 81.2 |  66.6 |
> |     | SupAP     | 64.1 | 81.9 |  67.4 |
> |     | SmoothCal | 64.0 | 82.0 |  68.9 |
> |     | ROADMAP   | **64.7** | **82.3** |  **69.3** |
> | 384 | SmoothAP  | 63.7 | 81.5 |  67.4 |
> |     | SupAP     | 64.6 | 81.9 |  68.4 |
> |     | SmoothCal | 63.8 | 81.5 |  69.0 |
> |     | ROADMAP   | **64.7** | **82.3** |  **69.2** |

---

> > ### Comment · Reviewer_9iMr · 2021-09-19
> > **Reviewer response**
> >
> > Thank you the authors for the response. I think they have provided adequate responses to all reviewers' concerns without the need to significantly edit the draft. The extra results on ROxf-RPar also add a significant impact on the landmark retrieval side of the community. Also, all reviewers agree that the theoretical workings in this paper provide good motivation for the proposed method. Hence, I will retain my initial positive rating of 6: Marginally above the acceptance threshold and recommend acceptance.

---

### Official Review · Reviewer_PwyW · 2021-07-16

**Rating:** 6
**Confidence:** 5

**Summary:**

This paper proposes a new method for deep image retrieval. The main contribution is a differentiable and decomposable loss function based on average precision loss.
The method shows superior performance and the theoretical seems correct.

**Main Review:**

Although the idea of differentiable AP loss is not a new concept, the solution given by the paper is novel. Especially the decomposable ap loss.

Both theoretical analysis and experimental justifications are given for the proposed method. Overall, the quality of the paper reaches the bar of the NeurIPS conference.

The paper is well written and the idea is clearly expressed. But the organization can be improved.

The smooth step function in Eq.(4)  weights the gradients better, however, this seems to be a trivial modification of the sigmoid function. Also, in Table 1, the comparison is unfair, since ROADMAP  also gains performance from the decomposable AP loss.

The comparison to other memory methods is an important experiment. The results should be included in the main text, rather than in the supplementary. (Also, typo in line 200, it should be Table 2). I suggest removing the qualitative results of Figure 6  to squeeze some space since it is not that indicative. It would be interesting to show how ROADMAP corrects some failing cases of other methods.




**Time Spent Reviewing:**

4

---

> ### Author Response · Authors · 2021-08-10
> **Author response to Reviewer 2**
>
> Thank you for the constructive and detailed feedback, and your kind words. We are happy to address your remarks and suggestions.
>
> **“The smooth step function in Eq.(4) weights the gradients better, however, this seems to be a trivial modification of the sigmoid function.”**
> *The modification is simple but has an important impact, which is thoroughly studied in the submission. Firstly, it provides a theoretical guarantee that the loss function $L_{\text{SupAP}}$ obtained from the smooth step function in Eq.(4)  is an upper bound on the non-differentiable AP loss, whereas $L_{\text{SmoothAP}}$ with the sigmoid is not. In terms of training, $L_{\text{SupAP}}$ has important algorithmic advantages over $L_{\text{SmoothAP}}$, especially its ability to prevent vanishing gradients, as shown in the gradient flow analysis in Figure 1 of the main paper (and further detailed in supplementary A).*
>
> **“Also, in Table 1, the comparison is unfair, since ROADMAP also gains performance from the decomposable AP loss.”**
> *The proposed ROADMAP loss in Eq (2) indeed includes two contributions compared to the AP optimization baselines shown in Table 1: the use of the $L_{\text{SupAP}}$  in Eq (5) and the $L_{\text{Calibr.}}$ loss in Eq (7). To further highlight the impact of each contribution, we provide ablation studies in Table 2 of the main paper (and Table 1 in Supplementary). This shows that $L_{\text{Calibr.}}$ is especially important when training with small batch size, while $L_{\text{SupAP}}$ still brings important gains for larger batch sizes.*
>
> **“The comparison to other memory methods is an important experiment. The results should be included in the main text, rather than in the supplementary.”**
> *We agree that this comparison is highly relevant. We will remove the qualitative results from the main paper and add the comparison to XBM in the main paper.*
>
> **“Also, typo in line 200, it should be Table 2”**
> *Thanks for your thorough reading, this is indeed a typo that will be corrected.*
>
> **“It would be interesting to show how ROADMAP corrects some failing cases of other methods.”**
> *In the paper we show quantitatively that ROADMAP produces a better ranking. We will add new qualitative results in the supplementary material to highlight the capacity of ROADMAP to correct failing cases of direct AP optimization competitors and memory methods.*

---

> > ### Comment · Reviewer_PwyW · 2021-08-31
> > **Good rebuttal**
> >
> > The authors did a good rebuttal and eased my concern. Also, I have checked the authors' responses to other reivewers, I am sure that the promised changes and new experimental justifications will make the paper more solid. Therefore, I will keep my initial rating.

---

### Official Review · Reviewer_mYow · 2021-07-17

**Rating:** 6
**Confidence:** 4

**Summary:**

The paper focuses on the problem of learning to rank. The paper proposes a method, ROADMAP (RObust And DecoMposable Average Precision), consisting of two main contributions:
- A new surrogate loss for average precision (AP). This loss is similar to recent losses that try to  provide a smooth approximation of AP / rank functions, but has some favorable design choices that lead to empirical improvements.
- A "calibration" loss (essentially a classification loss), to be used together with the AP loss, that implicitly provides consistency between batches and facilitates training on small batches.
Evaluation shows improvements with respect to recent AP approximations (e.g. [2, 3, 27]) on standard benchmarks.

**Limitations And Societal Impact:**

Paper describes societal impact in a reasonable manner in the conclusions of the paper.
Paper does not discuss technical limitations.

**Main Review:**

Novelty / Originality: the paper builds on recent AP approximation methods, particularly SmoothAP[2] (as seen clearly in Table 2). From that perspective, the paper is somewhat incremental. However, the modifications proposed are non-trivial and lead to improved results. I think the paper reaches the minimum level of novelty and originality required to be considered for acceptance.

Clarity: Mostly well written and easy to follow, although section 3.2 (Decomposable Average Precision) was, comparatively, not as clear as the rest of the paper. Perhaps the authors would like to revise that section if the paper gets accepted.

Method: The SupAP loss is well motivated, both from an intuitive and a theoretical point of view. The calibration loss is not so clear, and, although I see the main idea behind it (L_AP over batches underestimates L_AP, so a loss to reduce this gap is introduced. This is achieved by introducing a separation between positives and negatives), I have a some questions for the authors.

Evaluation: The method is tested on standard benchmarks (CUB, SOP, iNaturalist) and compared against recent AP approximations ([2, 3, 27, ...]) using recent architectures (ResNet, DeiT). Results show the superiority of the proposed approach on these benchmarks. Ablation studies are also shown.

Reproducibility: Code is not provided yet (authors mention that they plan to release it but don't have clearance yet). Hyperparameters and other technical details are provided. A domain expert should be able to reimplement the method with a reasonable chance of success.

Significance: this method follows a series of approaches that are becoming more and more popular in recent years. Given that, and given the strong empirical results, I'd say the paper's contribution is significant, although, as I mentioned before, I see it as somewhat incremental wrt [2].

Questions for authors:

Regarding the calibration loss: my understanding is that, ideally, one should reduce the gap between the AP on the whole dataset and the AP on a batch, and the proposed loss implicitly reduces that gap. However, doesn't this mean that the loss could actually reduce that gap by hurting both the global AP and the batch AP? is there any mechanism to prevent this (beyond the L_supap loss)?
[In general, I can't keep but think that the authors found that separating the positives and negatives within a batch (a very reasonable thing to try) had a positive impact, and built the calibration loss around this idea, hence the reason it feels convoluted.]

- From Table 2, could we see the results of using L_calib without using L_supap? I think this is important to disentangle the impact of each contribution.I've seen Figure 4a, but I think this information should be available in table 2 on all datasets. Also, why is map@R lower in table 2 than in figure 4a? Is the model in table 2 weaker? If that's the case I think we should see the ablation study on a strong model as well.

- There seems to be a divide in the ranking community between groups that report results in datasets like CUB or SOP, and groups that  report results in datasets such as Oxford and Paris. I am not requesting additional experiments during the review period. However, if the authors wished to add results on these datasets following standard training practices (see e.g. https://github.com/filipradenovic/cnnimageretrieval-pytorch for a codebase that allows training) either on an appendix of the final paper or a future technical report, I think this would considerably benefit the community.

Nit: What does the asterisk (*) denote by SoftBin in Table 3?

**Time Spent Reviewing:**

3

---

> ### Author Response · Authors · 2021-08-10
> **Author response to Reviewer 1**
>
> Thank you for reviewing our paper and your meaningful and valuable comments. We are happy to address your questions and remarks in a point-by-point response below.
>
> **“the calibration loss[...] reduce the gap between the AP on the whole dataset and the AP on a batch. However, doesn't this mean that the loss could actually reduce that gap by hurting both the global AP and the batch AP? is there any mechanism to prevent this (beyond the L_supap loss)?”**
> *This is an important point: the calibration loss $L_{\text{Calibr.}}$ does not hurt the batch AP or the AP on the whole dataset, it actually increases both. Indeed, the AP on a batch increases when the positives’ scores are larger than $\alpha$, and the negatives’ scores are lower than $\beta$ (with  $\alpha > \beta$). In addition, as you noted, the main goal of $L_{\text{Calibr.}}$ is to provide an absolute reference score for comparing ranking between batches in order to improve AP decomposability and to increase the AP on the whole dataset. The two losses $L_{\text{Calibr.}}$ and $L_{\text{SupAP}}$ are thus complementary ; $L_{\text{SupAP}}$ is a tighter upper bound of the batch AP than $L_{\text{Calibr.}}$, such that both losses cooperate towards the ultimate goal of maximizing the AP on the whole dataset.*
>
> **“From Table 2, could we see the results of using L_calib without using L_supap?”**
> *To fulfill this request, we report below the performances of $L_{\text{Calibr.}}$ without using $L_{\text{SupAP}}$ (i.e. on top of $L_{\text{SmoothAP}}$), that we denote as SmoothCal. The main outcome of this experiment is that $L_{\text{Calibr.}}$ and $L_{\text{SupAP}}$ are important in different regimes.
> Indeed, $L_{\text{SmoothCal}}$ enables large gains when the  (batch size) / (dataset size) ratio is small. For example, on INaturalist with batch size 64, $L_{\text{SmoothAP}}$ is 59.8 in R@1 vs 64.3 R@1 for $L_{\text{SmoothCal}}$ loss, which is close to the 64.5 of ROADMAP and superior to 61.2 for $L_{\text{SupAP}}$. On the same batch size, the trend is different on SOP, where $L_{\text{SmoothCal}}$ reaches similar scores (81.5 R@1) compared to $L_{\text{SupAP}}$ (81.4 R@1), and using ROADMAP further brings an important gain (82 R@1).
> For larger (batch size) / (dataset size) ratios, $L_{\text{SupAP}}$ can be equivalent and sometimes superior to $L_{\text{SmoothCal}}$, e.g. $L_{\text{SmoothCal}}$ 81.5 R@1 vs $L_{\text{SupAP}}$ 81.9 R@1 on SOP (batch size 384) or $L_{\text{SmoothCal}}$ 64.0 R@1 vs $L_{\text{SupAP}}$ 64.1 R@1 on CUB (batch size 224), with ROADMAP again further improving performances (64.7 R@1).
> We will be glad to add these results in the final paper if accepted.*
>
> Table 1: Ablation studies for the impact of our two contributions and the $L_{\text{SmoothAP}}$ baseline on the three datasets. Models are ResNet-50 trained with different batch sizes.
>
> | BS  | Loss      | CUB  | SOP  | INat  |
> |:-----:|:-----------|:------:|:------:|:-------:|
> | 64  | SmoothAP  | 62.1 | 81.0 |  59.8 |
> | | SupAP     | 62.9 | 81.4 |  61.2 |
> | | SmoothCal | 63.7 | 81.5 |  64.3 |
> |   | ROADMAP   | **63.8** | **82.0** |  **64.5** |
> | 224 | SmoothAP  | 62.9 | 81.2 |  66.6 |
> |  | SupAP     | 64.1 | 81.9 |  67.4 |
> |  | SmoothCal | 64.0 | 82.0 |  68.9 |
> |  | ROADMAP   |**64.7** | **82.3** |  **69.3** |
> | 384 | SmoothAP  | 63.7 | 81.5 |  67.4 |
> |  | SupAP     | 64.6 | 81.9 |  68.4 |
> |  | SmoothCal | 63.8 | 81.5 |  69.0 |
> |  | ROADMAP   | **64.7** | **82.3** |  **69.2**|
>
> **“Why is map@R lower in table 2 than in figure 4a? Is the model in table 2 weaker? If that's the case I think we should see the ablation study on a strong model as well.”**
> *The difference lies in the batch size: in Table 2, we use a batch size of 64 (see l. 188-189 in submission), while a batch size of 224 is used in Figure 4 (as indicated in Figure 4’s caption). Table 3 (and Table 4 for transformer backbones) in supplementary provides additional ablation studies using stronger models with larger batch sizes, showing the same trends as in Table 2 of the main paper. Note that the result in Table 3 of the supplementary for ROADMAP (27.74 in mAP@R) matches the one reported in Figure 4a for $\lambda = 0.5$.*
>
> **“There seems to be a divide in the ranking community between groups that report results in datasets like CUB or SOP, and groups that report results in datasets such as Oxford and Paris.”**
> *Since the submission deadline, we have run experiments on another dataset, InShop [a], on which we observed the same trends as in the 3 datasets of the main paper. Using the same experimental setup and only changing the training loss ROADMAP obtained 91.3 vs 90.1 in R@1 for SmoothAP. Again in a similar setup to the one used in [33], ROADMAP outperforms the state-of-the-art method ProxyNCA++ (91.3 vs 90.9  R@1).
> To fulfill R1’s request, we have run preliminary experiments to evaluate ROADMAP on $\mathcal{R}$Oxford and $\mathcal{R}$Paris, by training our model on the SfM-120k dataset and using the standard GitHub code for evaluation (https://github.com/filipradenovic/cnnimageretrieval-pytorch). With a similar transformer backbone as in [7], we obtained the following improvements shown below:*
>
> Table 2. Comparison of ROADMAP vs IRT [7] on $\mathcal{R}$Oxford and $\mathcal{R}$Paris. Models are DeiT-S [34], ROADMAP is trained with a batch size of 128.
>
> |          |  $\mathcal{R}$Ox M   | $\mathcal{R}$Ox H   | $\mathcal{R}$Par M   |  $\mathcal{R}$Par H   |
> |:---------|:------:|:------:|:------:|:------:|
> |IRT [7] | 34.5 | 15.8 | 65.8 | 42.0 |
> |ROADMAP| **38.9** | **20.7** | **67.5** | **42.3** |
>
> *We can see that ROADMAP is significantly better than [7] on $\mathcal{R}$Oxford and $\mathcal{R}$Paris medium protocol, and has similar performances for $\mathcal{R}$Paris hard protocol. This highlights the relevance of using ROADMAP instead of the contrastive loss used in [7].*
>
> *[a] Liu, Ziwei, et al. "Deepfashion: Powering robust clothes recognition and retrieval with rich annotations." Proceedings of the IEEE conference on computer vision and pattern recognition. 2016.*
>
> **“What does the asterisk (*) denote by SoftBin in Table 3?”**
> *This is to indicate that the reported results have been obtained by running experiments using the SoftBin method from the authors’ GitHub code (<https://github.com/naver/deep-image-retrieval/blob/master/dirtorch/loss.py>), since the results reported in the paper of Revaud et al. [27] do not include CUB, SOP or INaturalist datasets. We will make it clearer in the final version of the paper if accepted.*
>
> **“Reproducibility: Code is not provided yet (authors mention that they plan to release it but don't have clearance yet).”**
> *We obtained clearance to release the code from our industrial partner since the submission deadline. If accepted, we will thus post the code on GitHub with instructions on how to reproduce the results shown in the paper.*

---

> > ### Comment · Reviewer_mYow · 2021-09-02
> > **Thanks**
> >
> > Thanks for the answer, it addressed my concerns regarding technical and experimental aspects. Very glad to hear the code will be published if the paper is accepted.

---

### Decision · Program_Chairs · 2021-09-27

**Decision:**

Accept (Poster)

**Comment:**

The paper focuses on the problem of learning to rank and introduces a method, ROADMAP (RObust And DecoMposable Average Precision), consisting of a new surrogate loss for average precision (AP) and a "calibration" loss to be used together with the AP loss.

All reviewers recommended to accept.

Accept.